# Revisiting Meta-Learning with Noisy Labels: Reweighting Dynamics and Theoretical Guarantees

## Abstract

Learning with noisy labels remains challenging because over-parameterized networks memorize corrupted supervision. Meta-learning–based sample reweighting mitigates this by using a small clean subset to guide training, yet its behavior and training dynamics lack theoretical understanding. We provide a rigorous theoretical analysis of meta-reweighting under label noise and show that its training trajectory unfolds in three phases: (i) an alignment phase that amplifies examples consistent with a clean subset and suppresses conflicting ones; (ii) a filtering phase driving noisy example weights toward zero until the clean subset loss plateaus; and (iii) a post-filtering phase in which noise filtration becomes perturbation-sensitive. The mechanism is a similarity-weighted coupling between training and clean subset signals together with clean subset training loss contraction; in the post-filtering where the clean-subset loss is sufficiently small, the coupling term vanishes and meta-reweighting loses discriminatory power. Guided by this analysis, we propose a lightweight surrogate for meta reweighting that integrates mean-centering, row shifting, and label-signed modulation, yielding more stable performance while avoiding expensive bi-level optimization. Across synthetic and real noisy-label benchmarks, our method consistently outperforms strong reweighting/selection baselines.

## 1 Introduction

Driven by the rapid growth of accessible data, deep neural networks (DNNs) now achieve state-of-the-art performance across many tasks (He et al., 2016; Krizhevsky et al., 2012). Yet large datasets are often annotated by humans or scraped from the web, making labels inexpensive but error-prone in practice (Yu et al., 2018; Cheng et al., 2020). Such label noise—stemming from ambiguity, bias, or systematic mistakes—poses a serious challenge because modern DNNs are expressive enough to memorize arbitrary label assignments (Zhang et al., 2016). Consequently, even modest noise levels can mislead training and harm generalization through overfitting to corrupted supervision (Zhang et al., 2016; Arpit et al., 2017). Developing and understanding learning methods that remain reliable under noisy labels is therefore a central and urgent problem.

To curb memorization of corrupted labels, a growing literature develops methods for noisy label learning. Among existing strategies, sample reweighting and selection (i.e., "noise cleansing") are widely used for their strong empirical performance and compatibility with contrastive and unsupervised learning frameworks (Li et al., 2020; Miao et al., 2024). Sample reweighting/selection down-weights or filters suspicious examples using signals such as per-example loss (Gui et al., 2021), inter-model agreement (Wei et al., 2020), or cross-validation/large-loss heuristics (e.g., periodically partitioning data and removing high-loss instances) (Chen et al., 2019). Within the sample reweighting methods, meta-learning–based reweighting has emerged as a prominent approach: a small trusted validation set guides a bilevel optimization that learns example weights to improve downstream generalization (Ren et al., 2018; Shu et al., 2019; Holtz et al., 2022). However, scalability remains a key bottleneck since computing hypergradients via unrolling or implicit differentiation incurs substantial memory and compute overhead on modern architectures. Additionally, the theoretical underpinnings that drive the success of meta-learning-based reweighting are still incomplete. It is not yet fully understood when and why meta-reweighting provably suppresses noisy

examples, how the quality and bias of the validation set affects generalization, or how inner loop hyperparameters converge.

Existing convergence results for gradient-based meta-learning guarantee nonasymptotic convergence to first-order stationary points (FOSPs) (e.g., (Shu et al., 2019; Ji et al., 2022)). Such optimization guarantees alone do not preclude overfitting to noisy labels. In particular, Zhai et al. (2022) shows that, under overparameterization and standard losses, a broad class of sample-reweighting schemes inherits an implicit bias of empirical risk minimization (ERM) and may therefore offer no improvement over ERM. That analysis, however, assumes per-example weights converge to strictly positive limits and ignores the regularization effect introduced by the trusted meta clean set that guides meta-reweighting. These gaps require a sharper theory of meta-reweighting under label noise. In particular, we demonstrate that a careful analysis that models the interaction between weight dynamics and meta-set bias informs the design of more efficient and robust algorithms. A detailed discussion of related work is provided in Appendix A.

In this work, we analyze the training dynamics of meta-reweighting under noisy labels. We show that when the reweighting step size scales inversely proportional to the classifier training step size, the classifier is guaranteed to separate clean from noisy data. More concretely, under standard overparameterization and stable step sizes, training follows a three-phase trajectory: an early stage that boosts examples aligned with the clean subset signal while suppressing conflicting ones; a filtering stage during which the weights polarize so clean samples dominate, and which terminates when the clean subset error has converged (within a small tolerance); and a post-filtering stage where the signal naturally tapers and polarization becomes susceptible to perturbations. The effect is driven by a signed, similarity-weighted coupling between training and clean subset signals together with residual contraction. Our analysis clarifies when and how label filtering occurs and offers practical guidance for hyperparameters and the design of a particular training methodology. We demonstrate this by introducing a computationally light surrogate for the bilevel update. Specifically, our surrogate retains the signed, similarity-weighted aggregation, eliminates the need for backpropagation through inner-loop updates, and mean-centers the similarity Gram matrix to remove global bias. Empirically, on both synthetic and real-world datasets, the method consistently improves test accuracy over common baselines, demonstrating efficient noisy-label robust learning. We summarize our key contributions as follows:

- **Theoretical analysis.** We rigorously analyze the training dynamics of meta reweighting and characterize three stages. (1) Sample weight alignment stage where noisy weights decrease. (2) Noisy sample weights remain at zero while the loss of the clean subset converges. (3) The weights start to become perturbed and no longer filter out noisy samples.

- **Algorithmic surrogate.** Guided by our theory, we introduce a lightweight surrogate that updates sample weights using penultimate-layer feature similarity in place of the tangent-kernel similarity. To better align with the theoretical suggestions, we incorporate (i) mean-centering and row-shifting to remove global bias, (ii) label-signed modulation—i.e., we set the sign of each pairwise similarity by label agreement.

- **Empirical evaluation.** We empirically show that our method achieves higher accuracy than sample reweighting/selection baselines across multiple benchmarks and noise settings.

## 1.1 NOTATIONS

Let $\mathcal{X} \subset \mathbb{R}^d$ and $\mathcal{Y} \subset \mathbb{R}$ denote the input and output spaces, respectively, and assume $\|\mathbf{x}\|_2 \leq 1$ for all $\mathbf{x} \in \mathcal{X}$. We observe $\mathcal{D} = \{\mathbf{z}_i = (\mathbf{x}_i, y_i)\}_{i=1}^n$ drawn i.i.d. from $P$ over $\mathcal{X} \times \mathcal{Y}$. Let $\mathbf{X} = (\mathbf{x}_1, \ldots, \mathbf{x}_n) \in \mathbb{R}^{d \times n}$, the (hypothetical) clean labels $\mathbf{Y}^* = (y_1^*, \ldots, y_n^*)^\top \in \mathbb{R}^n$, and the observed (possibly noisy) labels $\mathbf{Y} = (y_1, \ldots, y_n)^\top \in \mathbb{R}^n$. In this work, we refer to any sample $\mathbf{x}_i$ with $y_i \neq y_i^*$ as a *noisy sample*, and to any sample with $y_i = y_i^*$ as a *clean sample*. For any mapping $g : \mathcal{X} \to \mathbb{R}^m$, we overload notation as $g(\mathbf{X}) = (g(\mathbf{x}_1), \ldots, g(\mathbf{x}_n)) \in \mathbb{R}^{m \times n}$ (a column vector when $m = 1$).

Let $f_\theta : \mathcal{X} \to \mathbb{R}$ be a neural network with parameters $\theta$ and loss $\ell : \mathbb{R} \times \mathcal{Y} \to \mathbb{R}_+$. Write $f_i(\theta) := f_\theta(\mathbf{x}_i)$ and the per-example loss $l_i(\theta) := \ell(f_\theta(\mathbf{x}_i), y_i)$. For nonnegative weights $w = (w_1, \ldots, w_n)^\top \in \mathbb{R}_{\geq 0}^n$, define the weighted empirical risk $L(\theta, w) := \frac{1}{n} \sum_{i=1}^n w_i \, l_i(\theta)$, which reduces to standard ERM when $w_i \equiv 1$. In the analysis we take $\mathcal{Y} = \{+1, -1\}$ and the squared loss

$\ell(\hat{y}, y) = \frac{1}{2}(\hat{y} - y)^2$ with $\hat{y} \in \mathbb{R}$ and $y \in \{+1, -1\}$. Let $u(\theta) = (u_1(\theta), \ldots, u_n(\theta))^\top \in \mathbb{R}^n$ denote residuals with $u_i(\theta) := f_\theta(\mathbf{x}_i) - y_i$, and let $J(\theta) := \nabla_\theta f_\theta(X) \in \mathbb{R}^{p \times n}$ denote the Jacobian. For the clean subset, we use $f_i^v(\theta)$, $u^v(\theta)$, and $J^v(\theta)$ analogously.

## 2 BACKGROUND AND META REWEIGHTING FRAMEWORK

There has been a growing line of work on *meta reweighting* for robust training. For example, Ren et al. (2018) propose a framework to learn per-example scalar weights by differentiating a validation objective through a one-step proxy of the training update, thereby steering optimization toward samples that improve held-out performance (e.g., under label noise or class imbalance). Shu et al. (2019) parametrize and learn a smooth weighting function by optimizing a meta-objective on a clean validation set, mapping example statistics (such as the loss suffered on each training example) to adaptive weights. Motivated by the implicit bilevel mechanism of these methods, we adopt a formulation that treats $w \in [0, 1]^n$ as an optimizable parameter. At each outer iteration $t$, we (i) update $w$ by performing one inner gradient steps on the validation objective evaluated at a one-step proxy $\widehat{\theta}_{t+1}(w)$ of the parameter update, and (ii) update $\theta$ once using the resulting weights $w_{t+1}$. This abstraction retains the essential mechanism of meta reweighting without method-specific design choices and serves as the basis for our analysis in the next sections.

**Pseudo update step.** At outer iteration $t$, given model parameters $\theta_t$ and example weights $w_t = (w_t^1, \ldots, w_t^n)$, we consider the following one-step proxy that maps weights to a tentative parameter update:

$$\widehat{\theta}_{t+1}(w) := \theta_t - \eta \nabla_\theta \Big( \sum_{i=1}^n w^i \, l_i(\theta_t) \Big),$$

where $l_i(\theta)$ denotes the training loss of example $i$ and $\eta > 0$ is the classifier learning rate. Using the clean subset losses $\{l_j^v(\cdot)\}_{j=1}^m$, define the meta objective as a function of the weights

$$\mathcal{L}_{\mathrm{val}}(w) := \frac{1}{m} \sum_{j=1}^m l_j^v(\widehat{\theta}_{t+1}(w)).$$

**Reweighting parameter update.** Given $w_t$, take one gradient step on $\mathcal{L}_{\mathrm{val}}$ with step size $\alpha > 0$:

$$\widetilde{w}_{t+1} = w_t - \alpha \nabla_w \mathcal{L}_{\mathrm{val}}(w_t).$$

Equivalently, unrolling the pseudo update step by the chain rule yields the following scheme:

$$w_{t+1}^i = w_t^i - \alpha \frac{\partial}{\partial w^i} \Big( \sum_{j=1}^m l_j^v(\theta_t - \eta \nabla_\theta \sum_{k=1}^n w_t^k l_k(\theta_t)) \Big), \qquad i = 1, \ldots, n. \tag{1}$$

In this work, we choose the reweighting stepsize as $\alpha = \beta/\eta$ with a fixed constant $\beta > 0$, so that the effective product $\alpha\eta$ remains constant; indeed, the $w$-update in Equation (1) depends on $\eta$ only through this product. We require the weights to lie in $[0, 1]$. Therefore, after each step we apply elementwise clipping:

$$w_{t+1}^i = \min\{1, \max\{0, \widetilde{w}_{t+1}^i\}\}, \qquad i = 1, \ldots, n.$$

**Classifier parameter update.** Using the updated weights $w_{t+1}$, perform one classifier update of the model parameters:

$$\theta_{t+1} = \theta_t - \eta \nabla_\theta \Big( \sum_{i=1}^n w_{t+1}^i \, l_i(\theta_t) \Big). \tag{2}$$

## 3 THEORETICAL ANALYSIS

In this section, we will study the learning dynamics of the meta reweighting algorithm in Section 2 under certain assumptions. Specifically, we will prove that at the early phase of training, the weights of noisy samples decrease to zero while weights of clean samples increase to one. Then, the weights will stay constant until the training loss of clean subset converges to $O(\eta + \widetilde{d}^{-1/4})$. Finally,

since the training loss of the clean subset is small, the weight gradient signal is overwhelmed by approximation errors. As a result, the weight update starts to be perturbed and noisy sample weights could be potentially positive.

We consider training dynamics of sufficiently wide fully-connected neural networks. Specifically, we define the output of the $l + 1$-th hidden layer of a $L$-layer neural network to be

$$\boldsymbol{h}^{l+1} = \frac{A^l}{\sqrt{d_l}} \boldsymbol{x}^l + \boldsymbol{b}^l \quad \text{and} \quad \boldsymbol{x}^{l+1} = \sigma\left(\boldsymbol{h}^{l+1}\right) \quad (l = 0, \cdots, L)$$

where $\sigma$ is a non-linear activation function, $A^l \in \mathbb{R}^{d_{l+1} \times d_l}$ and $\boldsymbol{b}^l \in \mathbb{R}^{d_{l+1}}$. Here $d_0 = d$. The parameter vector $\theta$ consists of $A^0, \cdots, A^L$ and $\boldsymbol{b}^0, \cdots, \boldsymbol{b}^L$ ( $\theta$ is the concatenation of all flattened weights and biases). The final output is $f(\boldsymbol{x}) = \boldsymbol{h}^{L+1}$. And let the neural network be initialized as

$$\begin{cases} A_{i,j}^l \sim \mathcal{N}(0,1) \\ b_j^l \sim \mathcal{N}(0,1) \end{cases} \quad (l = 0, \cdots, L-1) \quad \text{and} \quad \begin{cases} A_{i,j}^L = 0 \\ b_j^L = 0 \end{cases}$$

We also rely on the following assumption for the activation function:

**Assumption 1.** *$\sigma$ is differentiable everywhere. Both $\sigma$ and its first-order derivative $\dot{\sigma}$ are Lipschitz.*

**Wide Neural Networks:** The *neural tangent kernel* (NTK) of model $f_\theta$ is defined as $K^{(0)}(\boldsymbol{x}, \boldsymbol{x}') := \nabla_\theta f^{(0)}(\boldsymbol{x})^\top \nabla_\theta f^{(0)}(\boldsymbol{x}')$. We denote the Gram matrix by $\boldsymbol{K} := K^0(\boldsymbol{X}, \boldsymbol{X}) \in \mathbb{R}^{n \times n}$. When the width of $f_\theta$ increases to infinity, the NTK matrix converges to a deterministic kernel matrix:

**Lemma 1** (Zhai et al. (2022)). *If $\sigma$ is Lipschitz and $d_l \to \infty$ for $l = 1, \cdots, L$ sequentially, then $K^{(0)}(\boldsymbol{x}, \boldsymbol{x}')$ converges in probability to a deterministic limiting kernel $K(\boldsymbol{x}, \boldsymbol{x}')$ and the limiting kernel Gram matrix $\boldsymbol{K} := K(\boldsymbol{X}, \boldsymbol{X}) \in \mathbb{R}^{n \times n}$ is a positive definite symmetric matrix almost surely.*

We denote the largest and smallest eigenvalues of $\boldsymbol{K}$ by $\lambda_{\max}$ and $\lambda_{\min}$. Throughout this section, the network and data satisfy the following assumptions:

**Assumption 2.** *(i) $d_1 = \cdots = d_L = \widetilde{d}$; (ii) $\{\nabla_\theta f^{(0)}(\boldsymbol{x}_i)\}_{i=1}^n$ are linearly independent; and (iii) $\lambda_{\min} > 0$.*

For simplicity, we consider the data distribution support with the following property:

**Assumption 3.** *If $\boldsymbol{x}, \boldsymbol{x}'$ belongs to the same class, $K^{(0)}(\boldsymbol{x}, \boldsymbol{x}') > \gamma > 0$. Otherwise $K^{(0)}(\boldsymbol{x}, \boldsymbol{x}') < -\gamma < 0$.*

We will explain how this assumption can be relaxed in Section 3.2 and provide empirical evidence to demonstrate that this assumption and its relaxation are satisfied in practice.

### 3.1 LEARNING DYNAMICS

We now state our main result on the learning dynamics of the meta reweighting algorithm. Under the assumptions above and with an appropriately chosen learning rate (namely, $\eta \leq \widetilde{\eta}$ and reweighting step size scales as $\alpha\eta = \beta$), the trajectory exhibits, with high probability, a three-phase behavior: an *early phase* in which clean examples are upweighted and noisy examples are downweighted; a *filtering phase* in which the weights polarize to clean = 1 and noisy = 0 and the validation residual converges toward zero; and a *post-filtering phase* in which the residual-driven signal weakens and noisy sample weights are no longer guaranteed to be zero. The formal statement follows.

**Theorem 1.** *Let $f_{\theta_t}$ denote a fully connected neural network satisfying Assumptions 1 and 2. Suppose the data distribution satisfies Assumption 3. Then there exists $\widetilde{\eta} > 0$ such that, if $\eta \leq \widetilde{\eta}$ and $\alpha = \Omega(\eta^{-1})$, the following holds: for any $\delta > 0$ there exists $D > 0$ so that, whenever $\widetilde{d} \geq D$, with probability at least $1 - \delta$ over random initialization, there exist $0 < T_1 < T_2$ for some $T_1 = 1 + (m\alpha\eta\gamma)^{-1}$ such that:*

   1. *(Early phase) For $0 < t \leq T_1$, $w_t^i \in [1/2, 1]$ for all clean samples and $w_t^i \in [0, 1/2]$ for all noisy samples. Moreover, at $t = T_1$ the weights polarize to the extremes: $w_{T_1}^i = 1$ for all clean samples and $w_{T_1}^i = 0$ for all noisy samples.*

2. *(Filtering phase)* *For $T_1 < t \leq T_2$, $w_t^i$ remains zero on noisy samples; moreover,*

$$\|f_{\theta_{T_2}}(x^v) - y^v\|_\infty = O(\eta + \widetilde{d}^{-1/4}).$$

3. *(Post-filtering phase)* *For all $t \geq T_2$, the weights are no longer guaranteed to filter out the noisy samples.*

The proof is based on the first order approximation of update Equation (1). We can directly compute $\nabla_w \sum_{j=1}^m l_j^v(\theta_t - \eta \nabla \sum_{i=1}^n w_t^i l_i(\theta_t))$ via the chain rule:

$$\nabla_w \sum_{j=1}^m l_j^v(\theta_t - \eta \nabla \sum_{i=1}^n w_t^i l_i(\theta_t)) = -\eta U(\theta_t) J(\theta_t)^T J^v(\theta_t - \eta \nabla_\theta w_t^T l(\theta_t)) u^v(\theta_t - \eta \nabla_\theta w_t^T l(\theta_t))$$

$$= -\eta U(\theta_t) J(\theta_t)^T J^v(\theta_t) u^v(\theta_t) + e_1$$
$$= -\eta U(\theta_t) K(X, X_{\text{clean}}) u^v(\theta_t) + e_1 + e_2,$$

where $e_1 = O(\eta^2)$ is the truncation error resulting from first-order Taylor expansions of $J^v(\theta_t - \eta \nabla_\theta(w_t^\top \ell(\theta_t)))$ and $u^v(\theta_t - \eta \nabla_\theta(w_t^\top \ell(\theta_t)))$ around $\theta_t$, and $e_2$ denotes the discrepancy introduced by approximating $J(\theta_t)^\top J^v(\theta_t)$ with the NTK kernel $K(X, X_{\text{clean}})$.

At the beginning of training, the residuals $U(\theta_t)$ (and $u^v(\theta_t)$) are dominated by $\boldsymbol{y}$ (and $\boldsymbol{y}^v$) because of the zero-prediction initialization. Thus, the update direction of $\boldsymbol{w}$ is controlled by $\boldsymbol{y} K(X, X_{\text{clean}}) \boldsymbol{y}^v$. Under Assumption 3, weights of noisy (clean) samples will decrease (increase) according to the weight update formula. This dynamic remains true until the residuals are no longer controlled by $\boldsymbol{y}$ (and $\boldsymbol{y}^v$) or the weight update signal $U(\theta_t) K(X, X_{\text{clean}}) u^v(\theta_t)$ is smaller than the error term $e_1 + e_2$. Thus at the post-filtering phase, the weights start to be perturbed. A binary-MNIST experiment verifies this mechanism: as shown in Fig. 1a, the weights traverse the three phases described above; in the late stage, noisy-sample weights drift slightly from zero and evolve only slowly, which can lead to overfitting to label noise.

## 3.2 Data Distributions that are Well-Separated in Kernel Space

It is demonstrated in the proof of Theorem 1 that Assumption 3 is sufficient for our theoretical guarantees but not necessary. It requires the kernel to induce well-separated, label-consistent clusters: same-class pairs have positive tangent similarity, cross-class pairs have negative tangent similarity. We next investigate how our results extend under weaker conditions, relaxing Assumption 3.

**Shifted Kernel Values** We can replace Assumption 3 with the following:

**Assumption 4.** *There exists a constant $\mu$ such that if $\boldsymbol{x}, \boldsymbol{x}'$ belongs to the same class, $K^{(0)}(\boldsymbol{x}, \boldsymbol{x}') - \mu > \gamma > 0$. Otherwise $K^{(0)}(\boldsymbol{x}, \boldsymbol{x}') - \mu < -\gamma < 0$. Moreover, we assume $\frac{1}{m}|\sum_i u_i^v(\theta_t)| \lesssim \widetilde{d}^{-1/4}$.*

Under this relaxed assumption, the proof of the main result follows from the same idea. We empirically verify Assumption 4 by plotting the histogram of NTK values (Fig. 1c) and the mean of residuals over the clean meta subset (Fig. 1b).

**Mean-Centered Kernel Values** Let $K_{\boldsymbol{\mu}}$ denote the $\boldsymbol{\mu}$-centered kernel for some vector $\boldsymbol{\mu}$, then

$$u_i(\theta_t) K_{\boldsymbol{\mu}}(x_i, X_{\text{clean}}) u^v(\theta_t) = u_i(\theta_t) \left( \nabla_{\theta_t} f_{\theta_t}(x_i) - \boldsymbol{\mu} \right)^\top \left( \nabla_{\theta_t} f_{\theta_t}(X_{\text{clean}}) - \boldsymbol{\mu} \mathbf{1}_m^\top \right) u^v(\theta_t)$$

$$= u_i(\theta_t) \Big[ K(x_i, X_{\text{clean}}) u^v(\theta_t) - \nabla_{\theta_t} f_{\theta_t}(x_i)^\top \boldsymbol{\mu} \left( \mathbf{1}_m^\top u^v(\theta_t) \right)$$

$$- \boldsymbol{\mu}^\top \nabla_{\theta_t} f_{\theta_t}(X_{\text{clean}}) u^v(\theta_t) + \boldsymbol{\mu}^\top \boldsymbol{\mu} \left( \mathbf{1}_m^\top u^v(\theta_t) \right) \Big] \qquad (3)$$

Since the clean subset is balanced, we further obtain from equation 3 that

$$u_i(\theta_t) K_\mu(x_i, X_{\text{clean}}) u^v(\theta_t) = u_i(\theta_t) \Big( K(x_i, X_{\text{clean}}) u^v(\theta_t) + c_0 \Big)$$

where $c_0 = -\boldsymbol{\mu}^\top \nabla_{\theta_t} f_{\theta_t}(X_{\text{clean}}) u^v(\theta_t)$, which is independent of the individual sample $x_i$. Since $u_i = -y_i$ at the beginning of training, the weight update direction is thus $y_i K(x_i, X_{\text{clean}}) \boldsymbol{y}^v - y_i c_0$.

---

**Algorithm 1** Feature-Based Reweighting (FBR): a lightweight surrogate for meta reweighting

---

**Input:** Training set $\{(x_i, y_i)\}_{i=1}^n$, clean subset $\{(x_j^v, y_j^v)\}_{j=1}^m$, feature map $\phi : \mathcal{X} \to \mathbb{R}^d$, stepsize $\alpha > 0$, label-signed coefficients $(\lambda_+, \lambda_-) \geq 0$, iterations $T$.
**Output:** Weights $\boldsymbol{w}_T$, network $f_{\theta_T}$.

1: Initialize $w_i^0 = \frac{1}{2}$ for all $i$.
2: **for** epoch $t = 0, \cdots, T-1$ **do**
3:      $\mu \leftarrow \frac{1}{m} \sum_{j=1}^m \phi(x_j^v)$                                                   $\triangleright$ clean-set mean
4:      $\widetilde{\phi}(x_j^v) \leftarrow \phi(x_j^v) - \mu$ for $j = 1, \ldots, m$;      $\widetilde{\Phi}_v \leftarrow [\widetilde{\phi}(x_j^v)]_{j=1}^m$
5:      **for** mini-batch $B \subseteq \{1, \ldots, n\}$ **do**
6:          $\widetilde{\phi}(x_i) \leftarrow \phi(x_i) - \mu$ for all $i \in B$;      $\widetilde{\Phi}_B \leftarrow [\widetilde{\phi}(x_i)]_{i \in B}$
7:          $K_B \leftarrow \widetilde{\Phi}_B \widetilde{\Phi}_v^\top$                                  $\triangleright |B| \times m$ mean-centered Gram
8:          **for** each $i \in B$ **do**
9:              $s_{i,c} \leftarrow \frac{1}{|\{j : y_j^v = c\}|} \sum_{j : y_j^v = c} (K_B)_{i,j}$ for $c = 1, \ldots, C$
10:            $c_i^{(1)} \leftarrow \arg\max_c s_{i,c}$;    $c_i^{(2)} \leftarrow \arg\max_{c \neq c_i^{(1)}} s_{i,c}$         $\triangleright$ top-1 / top-2 means
11:            $\widetilde{K}_{i,j} \leftarrow (K_B)_{i,j} - s_{i, c_i^{(2)}}$ for $j = 1, \ldots, m$             $\triangleright$ row shifting
12:            $\bar{K}_{i,j} \leftarrow \big(\lambda_+ \mathbf{1}\{y_i = y_j^v\} - \lambda_- \mathbf{1}\{y_i \neq y_j^v\}\big) \widetilde{K}_{i,j}$ for $j = 1, \ldots, m$
13:            $\widetilde{d}_i(\theta_t) \leftarrow \sum_{j=1}^m \bar{K}_{i,j}$                              $\triangleright$ row-sum direction
14:            $w_i^{t+1} \leftarrow \text{clip}_{[0,1]}(w_i^t - \alpha\, \widetilde{d}_i(\theta_t))$
15:          $\mathcal{L}_B \leftarrow \frac{1}{|B|} \sum_{i \in B} w_i^t \cdot \ell\big(f_{\theta_t}(x_i), y_i\big)$
16:          $\theta_{t+1} \leftarrow \theta_t - \eta \nabla_\theta \mathcal{L}_B$

**Proposition 1.** *Let $X_{\text{clean}} = \{x_j^{\text{clean}}\}_{j=1}^m$ be i.i.d. samples from a distribution $\mathcal{D}$. Let $K$ be an NTK kernel and assume it is bounded: $|K(x, y)| \leq K_{\max}$ for all $x, y$. Assume non-degeneracy: for any fixed $x_i$,*

$$\text{Var}_{X' \sim \mathcal{D}}\big[K(x_i, X')\big] \geq \sigma^2 > 0.$$

*Let the clean labels be balanced and set $\boldsymbol{u}^v = -\boldsymbol{y}^v \in \{\pm 1\}^m$ so that $\mathbf{1}_m^\top u^v = 0$ and $\|\boldsymbol{u}^v\|_2 = \sqrt{m}$. Write $K_m := K(X_{\text{clean}}, X_{\text{clean}}) \in \mathbb{R}^{m \times m}$ and suppose there exists (possibly random, i.e. data-dependent) $c_m \in \mathbb{R}$ such that*

$$\|K_m \mathbf{1}_m - c_m \mathbf{1}_m\|_2 = O_p\big(m^{1-\varepsilon}\big) \qquad \text{for some } \varepsilon > 0. \tag{4}$$

*Define*

$$c_0 := -\frac{1}{m} \mathbf{1}_m^\top K_m \boldsymbol{u}^v, \qquad S_i := K(x_i, X_{\text{clean}})\, \boldsymbol{u}^v.$$

*Then, as $m \to \infty$,*

$$|c_0| = O_p\big(m^{\frac{1}{2}-\varepsilon}\big), \qquad |S_i| = \Omega_p(\sqrt{m}), \qquad \frac{|c_0|}{|S_i|} = O_p\big(m^{-\varepsilon/2}\big) \xrightarrow{p} 0.$$

Proposition 1 implies that, for the first step analysis, Assumption 3 can be replaced with the same assumption on the corresponding mean-centered kernel and the row-sum regularity condition 4.

To validate the kernel value distribution assumption for the mean-centered kernel, we conduct experiments on a binary MNIST task and plot the value histogram during training in Fig. 1d. We also plot the histogram of the NTK values in Fig. 1c.

## 4 REVISITING META LEARNING-BASED ALGORITHMS

Our theoretical analysis indicates that the effectiveness of meta reweighting is governed by the *weight–update direction*

$$d_i(\theta_t) = u_i(\theta_t) \big[K(x_i, X_{\text{clean}})\, u^v(\theta_t)\big], \qquad i = 1, \ldots, n.$$

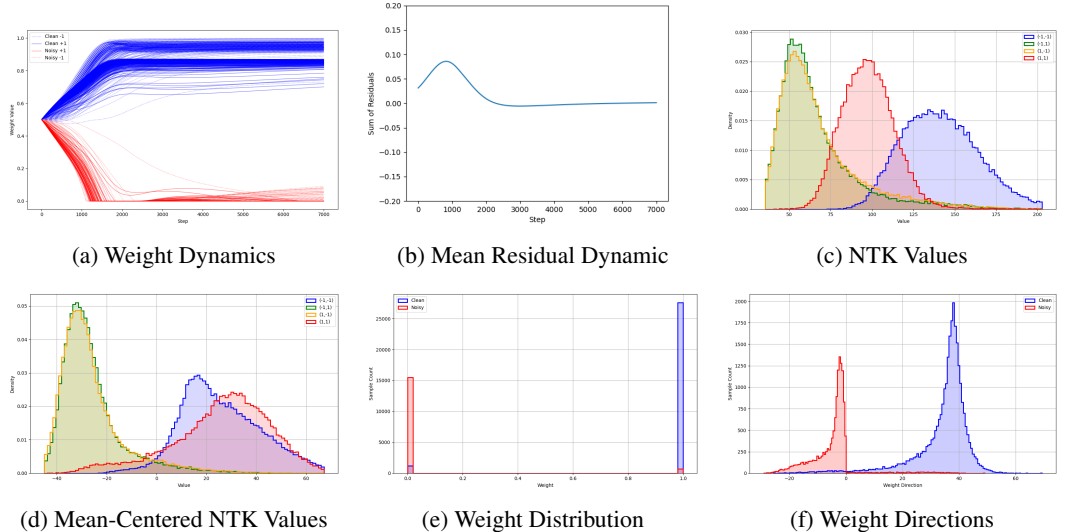

| (a) Weight Dynamics | (b) Mean Residual Dynamic | (c) NTK Values |
| --- | --- | --- |
| (d) Mean-Centered NTK Values | (e) Weight Distribution | (f) Weight Directions |

Figure 1: (a) Trajectories of per-sample weights over epochs. (b) Mean residual dynamics on the clean subset. (c) NTK values between training and clean samples. (d) Mean-centered NTK values between training and clean samples. (e) Final weight distribution at convergence, separated by clean vs. noisy samples. (f) Final weight directions.

This expression aggregates, for each training example $x_i$, the similarity between $x_i$ and the clean subset $X_{\text{clean}}$ weighted by the meta signals $u_i(\theta_t)$ and $u^v(\theta_t)$. As established in Theorem 1, if $u^v(\theta_t)$ decays toward zero, the useful update signal can be dominated by approximation error; moreover, even under relaxed distributional assumptions (cf. Assumption 4 and Proposition 1), the direction is susceptible to perturbations when the corresponding constants are large or $|X_{\text{clean}}|$ is small. In addition, the bilevel structure entails nontrivial computational overhead. Motivated by these observations, we propose a streamlined procedure for the multiclass setting that preserves the analytical structure of $u(\theta_t)K(\cdot, X_{\text{clean}})u^v(\theta_t)$ while avoiding heavy nested differentiation.

**(1) Mean-centered features.** Let $\phi : \mathcal{X} \to \mathbb{R}^d$ be a representation mapping, e.g., the penultimate-layer feature or the neural tangent feature. Define the clean-set mean $\mu := \frac{1}{m}\sum_{j=1}^{m}\phi(x_j^v)$ and the centered features $\widetilde{\phi}(x) := \phi(x) - \mu$. We then form a Gram matrix between the training and clean sets by

$$K(x_i, x_j^v) := \langle \widetilde{\phi}(x_i), \widetilde{\phi}(x_j^v)\rangle, \quad \text{equivalently} \quad K = \widetilde{\Phi}\,\widetilde{\Phi}_v^\top,$$

where $\widetilde{\Phi} \in \mathbb{R}^{n \times d}$ and $\widetilde{\Phi}_v \in \mathbb{R}^{m \times d}$ stack the centered features. Mean-centering removes the global bias discussed in Section 3.2. As indicated in our theoretical analysis, the Gram matrix entries are expected to be well-separated by class: within-class similarities exceed cross-class ones.

**(2) Multiclass row shifting.** Let $C$ be the number of classes. For each point $i$ and class $c$, compute class-wise similarity aggregates

$$s_{i,c} := \frac{1}{|\{j : y_j^v = c\}|} \sum_{j : y_j^v = c} K(x_i, x_j^v), \qquad c \in \{1, \ldots, C\},$$

and let $c_i^\star$ be the class with the second-largest mean similarity $s_{i,c}$. Define a row shift that preserves only the dominant class as positive:

$$\widetilde{K}_{i,j} := K_{i,j} - s_{i,c_i^\star}.$$

This step produces a class-discriminative margin at the row level without modifying the dominant class ordering.

**(3) Label-aware scaling.** To further align the update with the putative label of $x_i$, apply a label-aware multiplicative mask with $\lambda_+, \lambda_- \geq 0$:

$$\bar{K}_{i,j} := \left(\lambda_+ \mathbb{1}\{y_i = y_j^v\} - \lambda_- \mathbb{1}\{y_i \neq y_j^v\}\right)\widetilde{K}_{i,j},$$

where $\mathbb{1}\{\cdot\}$ is the indicator function. This operation can be viewed as a simple surrogate for $u_i(\theta_t)u^v(\theta_t)$-weighted alignment when reliable validation signals are weak or noisy.

**(4) Row-sum weight update and constraints.** Define the simplified direction

$$\widetilde{d}_i(\theta_t) \; := \; \sum_{j=1}^{m} \bar{K}_{i,j},$$

and update

$$w_{t+1}^i \; = \; \Pi_{[0,1]}\big(w_t^i - \alpha\,\widetilde{d}_i(\theta_t)\big), \qquad i = 1,\dots,n,$$

where $\Pi_{[0,1]}$ denotes elementwise clipping. This preserves the signed aggregation structure of the ideal meta direction while replacing the NTK with a centered computationally friendly surrogate.

**Remarks on computation and stability.** (i) The dominant cost is the matrix product $K = \widetilde{\Phi}\,\widetilde{\Phi}_v^{\top}$, which can be computed in $O(Bmd)$ time and $O(\min\{Bm,\, Bd + md\})$ memory. (ii) Mean-centering makes $K$ invariant to global feature shifts and mitigates the constant-component bias; this mirrors the role of kernel centering in our analysis. (iii) The row-shift construction ensures a nonpositive sum for all non-dominant classes. (iv) Hyperparameters $(\lambda_+, \lambda_-)$ control the within-/cross-class tradeoff; a practical default is $\lambda_+ = 1$ and $\lambda_- = 1/(C-1)$ with annealing.

In summary, the proposed construction yields a computationally friendly surrogate for the meta reweighting update $u(\theta_t)K(\cdot, X_{\text{clean}})u^v(\theta_t)$. This surrogate retains the essential signed, similarity-weighted aggregation motivated theoretically, while replacing the NTK with a centered Gram matrix and enforcing a multiclass similarity margin via row shifting and label-aware scaling. Computational details are provided in Algorithm 1.

**Distribution of weight derivatives.** On CIFAR-10 with $40\%$ symmetric noise, we run Algorithm 1 and record throughout training the *negative* weight derivatives $-\partial\mathcal{L}_{\text{val}}/\partial w_i$ (the instantaneous meta–update directions for $w_i$). The empirical distribution is shown in Fig. 1f. Consistent with the theory, the two populations are clearly separated: most clean examples yield positive values (driving $w_i$ upward), whereas most noisy examples yield negative values (driving $w_i$ downward). This discriminative signal underlies the subsequent filtering effect; the terminal weight histogram in Fig. 1e further affirms this separation.

**Remark 1** (Connection between binary MSE analysis and our proposed algorithm). *Our algorithm can be viewed as decoupling sample reweighting from the subsequent training step. In the theoretical analysis (Section 3), we consider a binary classifier trained with the MSE loss and show that the meta-reweighting direction can be expressed as a signed similarity between a training example and the clean subset via the NTK. In the multiclass setting, we lift this mechanism via a standard one-vs-all construction: for a sample $(x_i, y_i = c)$ and class $c$, we define an auxiliary binary problem by assigning label $+1$ to class $c$ and $-1$ to all other classes, and let $f_c(\theta, x)$ denote the logit for class $c$. Then the NTK-induced feature inner products $\left\langle \nabla_\theta f_c(\theta, x_i), \nabla_\theta f_c(\theta, X_{clean}) \right\rangle$ induces a kernel-based alignment between $(x_i, y_i)$ and the clean subset, which we use to build our reweighting signal. These per-example weights are subsequently applied in an arbitrary supervised training loss, including standard multiclass cross-entropy.*

*For this reweighting signal to be effective, it suffices that the kernel values preserve a certain degree of separability between clean and noisy samples throughout training. The NTK regime together with Assumption 3 (or other kernel separability assumptions) provides a* sufficient *condition under which a good reweighting signal is maintained over training. In more complicated settings, the kernel values may not remain constant if the neural network does not satisfy the NTK assumptions or if the training dynamics are driven by cross-entropy loss rather than the squared loss considered in our theory. Our experiments in Appendix D.10 show that this separability property persists empirically even when the kernel dynamics deviate from the idealized NTK regime or when the model is trained with multiclass cross-entropy.*

## 5 EXPERIMENTS

In this section, we empirically evaluate the effectiveness of our algorithm on both synthetic (i.e. CIFAR-10 and CIFAR-100) and realistic (i.e., CIFAR-N and Clothing-1M) label noise.

Table 1: Results on CIFAR-10/100 with symmetric and asymmetric label noise.

| Dataset | CIFAR-10 | | | CIFAR-100 | | |
|---|---|---|---|---|---|---|
| **Noisy Type** | Sym | | Asym | Sym | | Asym |
| **Noise Ratio** | 20 | 50 | 40 | 20 | 50 | 40 |
| Standard | $87.0 \pm 0.1$ | $78.2 \pm 0.8$ | $85.0 \pm 0.0$ | $58.7 \pm 0.3$ | $42.5 \pm 0.3$ | $42.7 \pm 0.6$ |
| Bootstrap | $86.2 \pm 0.2$ | – | $81.2 \pm 1.5$ | $58.3 \pm 0.2$ | – | $45.1 \pm 0.6$ |
| Forward | $88.0 \pm 0.4$ | – | $83.6 \pm 0.6$ | $39.2 \pm 2.6$ | – | $34.4 \pm 1.9$ |
| Co-teaching | $89.3 \pm 0.3$ | $83.3 \pm 0.6$ | $88.4 \pm 2.8$ | $63.4 \pm 0.4$ | $49.1 \pm 0.4$ | $47.7 \pm 1.2$ |
| Co-teaching+ | $89.1 \pm 0.5$ | $84.9 \pm 0.4$ | $86.5 \pm 1.2$ | $59.2 \pm 0.4$ | $47.1 \pm 0.3$ | $44.7 \pm 0.6$ |
| TopoFilter | $90.4 \pm 0.2$ | $86.8 \pm 0.3$ | $87.5 \pm 0.4$ | $66.9 \pm 0.4$ | $53.4 \pm 1.8$ | $56.6 \pm 0.5$ |
| CRUST | $89.4 \pm 0.2$ | $87.0 \pm 0.1$ | $82.4 \pm 0.0$ | $69.3 \pm 0.2$ | $62.3 \pm 0.2$ | $56.1 \pm 0.5$ |
| FINE | $91.0 \pm 0.1$ | $\mathbf{87.3 \pm 0.2}$ | $89.5 \pm 0.1$ | $70.3 \pm 0.2$ | $64.2 \pm 0.5$ | $61.7 \pm 1.0$ |
| BHN | $85.7 \pm 0.2$ | $85.4 \pm 0.1$ | $86.4 \pm 0.9$ | $64.1 \pm 0.3$ | $48.2 \pm 0.7$ | $53.4 \pm 0.9$ |
| RENT | $79.8 \pm 0.2$ | $66.8 \pm 0.6$ | $78.4 \pm 0.3$ | $48.4 \pm 0.6$ | $37.2 \pm 1.2$ | $37.8 \pm 1.4$ |
| **Ours (FBR)** | $\mathbf{92.3 \pm 0.2}$ | $87.0 \pm 0.1$ | $\mathbf{90.6 \pm 0.4}$ | $73.4 \pm 0.2$ | $\mathbf{65.4 \pm 0.4}$ | $\mathbf{73.2 \pm 0.2}$ |
| **Ours (NTK)** | $91.4 \pm 0.2$ | $86.4 \pm 0.2$ | $89.7 \pm 0.4$ | $\mathbf{73.6 \pm 0.2}$ | $\mathbf{65.4 \pm 0.5}$ | $73.1 \pm 0.4$ |

Table 2: Test accuracy on Clothing1M dataset.

| Method | Standard | SCE | ELR | CORES$^2$ | FINE | BHN | RENT | Ours |
|---|---|---|---|---|---|---|---|---|
| Accuracy | 68.94 | 71.02 | 72.87 | 73.24 | 72.91 | 73.27 | 70.1 | **74.16** |

**Experimental setup.** We adhere to the evaluation setup of FINE (Kim et al., 2021).

*Benchmarks and noise models.* Following standard practice, we consider two synthetic label-noise regimes on CIFAR-10/100: (i) *symmetric* noise, where each label is independently flipped to a uniformly random incorrect class at a prescribed rate; and (ii) *asymmetric* noise, with class-dependent flips. For CIFAR-10 we use the canonical mapping TRUCK→AUTOMOBILE, BIRD→AIRPLANE, DEER→HORSE, CAT↔DOG. For CIFAR-100, the 100 classes are grouped into 20 super-classes (size 5) and labels are shifted cyclically within each super-class. We also evaluate on the real-world Clothing1M dataset (Xiao et al., 2015) with naturally corrupted labels (1M images, 14 categories), which provides 50k/14k/10k verified-clean splits for training/validation/testing. As in (Kim et al., 2021), instead of using the 50k clean training split, we construct a pseudo-balanced training set of 120k images from the noisy pool and report accuracy on the 10k clean test set. For every dataset, we reserve a clean meta subset of 2,000 examples for weight updates. On CIFAR-10/100 this subset is drawn from the original clean labels; on Clothing1M it is drawn from the provided 14k clean validation set. We compare the proposed method with the following sample selection/reweighting baselines: (1) Bootstrap (Reed et al., 2014), (2) Forward (Patrini et al., 2017), (3) Co-teaching (Han et al., 2018), (4) Co-teaching++ (Yu et al., 2019), (5) TopoFilter (Wu et al., 2020), (6) CRUST (Mirzasoleiman et al., 2020), (7)BHN (Yu et al., 2023), (8) RENT (Bae et al., 2024). In the appendix, we expand our analysis to include the CIFAR-10/100-N dataset (Wei et al., 2022).

*Results overview.* Table 1 reports test accuracies across noise rates on CIFAR-10/100. On CIFAR-10 with 20% symmetric noise our method attains 92.3%: +1.3% gain over the strongest prior (FINE at 91.0%). At 50% symmetric noise we obtain 87.0%, which matches the performance of CRUST (87.0%) and is within the margin error compared to FINE (87.3%). At 40% asymmetric noise we reach 90.6% accuracy, improving on FINE (89.5%) by +1.1%. Gains are larger on CIFAR-100: at 20%/50% symmetric noise we achieve 73.4% and 65.4% respectively, exceeding FINE (70.3%, 64.2%) 3.0% and 1.2%, respectively. With 40% asymmetric noise we observe 73.2% accuracy, a more than 11% improvement over the best prior report (FINE at 61.7%). These trends indicate that our selection mechanism is especially effective in the class-dependent corruption regime and for the fine-grained CIFAR-100 label space, where overfitting to noise is most severe. On Clothing1M, our approach achieves 74.16% top-1 accuracy, surpassing the baselines. Empirically, the procedure retains informative clean instances while suppressing misleading ones, mitigating overfitting to noisy labels and yielding superior overall performance. In Table 3 in Appendix D.2, we report test accuracies for the CIFAR-N datasets. On CIFAR-10N Aggre our method attains 92.3% accuracy: a 0.9%

improvement over the strongest prior (JoCoR at 91.4%). On CIFAR-10N Worst, our method attains 85.6% accuracy: a 1.8% improvement over the best performing prior (Co-Teaching at 83.3%). Similar to the synthetic benchmarks, the improvement is more pronounced on CIFAR-100N (a +3.5% improvement over CORES).

We additionally compare with Meta-Weight-Net (MW-Net) (Shu et al., 2019) on CIFAR-10/100. Table 4 in Appendix D demonstrates the advantages of neural-tangent features and penultimate-layer features. In contrast, MW-Net shows marked overfitting in the presence of label noise.

## 6    CONCLUSIONS

We analyze the training dynamics of meta-reweighting under noisy labels when the reweighting step size scales inversely with the classifier training step size, and develop a theory explaining why the procedure separates clean from noisy data. Under standard over-parameterization and stable step sizes, training follows a three-phase trajectory—early alignment, filtering with weight polarization and clean-subset convergence, and post-filtering susceptibility. We further provide empirical evidence that corroborates both the predicted dynamics and the underlying assumptions.

Building on these insights, we proposed a computationally light surrogate for the bilevel update that preserves the same mechanism: it keeps the signed, similarity-weighted aggregation and mean-centers the similarity Gram matrix to remove global bias. Across synthetic and real-world datasets, the surrogate consistently improves test accuracy over strong reweighting/selection baselines.

## 7 REPRODUCIBILITY STATEMENT

We have taken several steps to facilitate reproduction of our results. An anonymous code package is provided in the supplementary materials. Experimental settings—datasets, preprocessing, model architectures, hyperparameters and training schedules are summarized in Section 5 and Appendix D. Theoretical results are stated with explicit assumptions and accompanied by complete proofs in Section 3 (see also Appendix B).

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

## A   PREVIOUS STUDIES

### A.1   THEORETICAL WORKS FOR META LEARNING

**Convergence analysis.**   For gradient-based meta-learning (GBML) such as MAML, several works establish convergence to an $\varepsilon$–first-order stationary point under appropriate task/data batch sizes and

step-size choices (Fallah et al., 2020; Ji et al., 2022). Moreover, using first-order updates that drop Hessian terms may forfeit these guarantees in general, while Hessian-free variants can recover them (Fallah et al., 2020). Multi-step MAML admits provable convergence in both resampling (fresh data per inner step) and finite-sum (fixed per-task dataset) regimes, with theory recommending inner-loop step sizes that shrink with the number of inner steps to control nested-gradient bias and variance (Ji et al., 2022).

**Generalization and benign overfitting.** Stability- and distributional-analyses yield task-level generalization bounds: for *recurring* test tasks (drawn from those seen in training), the excess risk decreases with both the number of tasks and per-task samples; for *unseen* tasks, the bound degrades with the discrepancy between the train-task and test-task distributions (e.g., via total-variation or related divergences) (Fallah et al., 2021). In over-parameterized meta-linear settings, recent results show that GBML can exhibit *benign overfitting*: the excess risk decomposes into cross-task and within-task variance plus bias terms, identifying regimes where interpolation does not harm—and can even improve—generalization; the theory highlights the role of task heterogeneity as well as inner-loop step size and regularization in maintaining this benign behavior (Chen et al., 2022).

## A.2 NOISY LABEL LEARNING

**Noisy-label learning (NLL).** Deep networks easily memorize label noise, hurting generalization even when training error goes to zero (Zhang et al., 2021). Prior work can be organized into three main directions—label correction, sample selection, and sample re-weighting—along with surveys that systematize these trends (Song et al., 2022).

*Sample re-weighting.* Re-weighting adjusts per-sample importance to mitigate corrupted labels (and other defects). Classical loss re-weighting (Liu & Tao, 2015) is complemented by meta-learning approaches that learn example weights from a small trusted set (Ren et al., 2018) or learn a parametric weighting function (Meta-Weight-Net) (Shu et al., 2019); these typically outperform heuristic weighting under label noise (Ren et al., 2018; Shu et al., 2019). Other strategies infer Bayesian latent weights (Wang et al., 2017), learn curricula via a mentor network (Jiang et al., 2018), or adopt self-paced updates based on loss (Kumar et al., 2010); analogous ideas appear in adversarial robustness (Holtz et al., 2022).

*Sample selection.* Sample selection can be regarded as a special case of sample reweighting. These methods filter or down-weight likely-noisy instances during training. The small-loss principle assumes clean samples have lower losses early on (Han et al., 2018), but may bias toward easy examples. To reduce confirmation bias and over-pruning, variants select by peer disagreement (Yu et al., 2019), co-regularize two networks with agreement (e.g., KL) (Wei et al., 2020), or adopt dynamic thresholds with confidence regularization (Cheng et al., 2020).

*Label correction.* Correction is often combined with *sample selection* to improve robustness and stability, with selection gating updates to likely-clean examples and correction amending mislabeled ones. Leveraging the early-learning effect—models correctly predict a subset of mislabeled samples early in training (Liu et al., 2020)—labels are corrected via (i) joint optimization of network parameters and (soft) labels (Tanaka et al., 2018) or (ii) probabilistic label modeling (Yi & Wu, 2019). Dual-network schemes stabilize correction (Co-Teaching exchanges small-loss examples (Han et al., 2018); SELFIE replaces a subset using historical predictions (Song et al., 2019)). Hybrid methods combine correction with *sample selection*, e.g., DivideMix uses a two-component mixture to split clean/noisy data and applies semi-supervised training with Mixup (Li et al., 2020; Zhang et al., 2017), which helps gate updates and curb confirmation bias. Additional criteria/schedules include likelihood-ratio relabeling (Zheng et al., 2020) and stage-/confidence-aware regularization (ProSelfLC) (Wang et al., 2021).

More recent work further integrates sample selection and label correction into sophisticated semi-supervised pipelines. Yuan et al. (Yuan et al., 2024) detect memorized noisy samples and exclude them from the supervised set, coupling this with a noise-aware MixMatch and a modified DivideMix-style framework. L2B (Zhou et al., 2024) employs a meta-learning framework to bootstrap robust models by jointly reweighting instances and interpolating between observed and pseudo labels within a semi-supervised relabeling scheme. DCD (Mu et al., 2025) meta-learns a dynamic center-distance metric for hard-sample mining, combining center-based selection and

label correction with an SSL training pipeline. Taken together, these methods form complex selection–relabeling–SSL systems that are complementary to our simpler meta-reweighting mechanism, which focuses on learning per-example weights from a small clean subset rather than designing a full semi-supervised training framework.

# B PROOF OF THEOREM 1

## B.1 EARLY PHASE

To prove Theorem 1, we first state the NTK result of wide neural networks.

**Definition 1.** *Let the linearized neural network of $f_{\theta_t}(x)$ be defined as $f_{\text{lin}}^{(t)}(\boldsymbol{x}) := f^{(0)}(\boldsymbol{x}) + \langle \theta^{(t)} - \theta^{(0)}, \nabla_\theta f^{(0)}(\boldsymbol{x}) \rangle$.*

**Theorem 2.** *Let $f_{\text{lin}}^{(t)}$ be its linearized neural network trained by the same reweighting factors. Under the same settings as in Theorem 1, for any fixed finite $T > 0$, with probability at least $1 - \delta$ over random initialization we have*

$$\left| f_{\text{lin}}^{(t)}(\boldsymbol{x}) - f^{(t)}(\boldsymbol{x}) \right| \leq C\eta t B^{t-1} \tilde{d}^{-1/4}.$$

Begin with the following Lemma:

**Lemma 2.** *There exist constants $M > 0$ such that for any $\delta > 0$, there exist $R_0 > 0, \tilde{D} > 0$ and $B > 1$ such that for any $\tilde{d} \geq \tilde{D}$, the following hold with probability at least $(1 - \delta)$ over random initialization and with learning rate $\eta$ : For all $t \leq T_1$, there is*

$$\left\| u\left(\theta^{(t)}\right) \right\|_2 \leq B^t R_0 \tag{5}$$

$$\sum_{j=1}^{t} \left\| \theta^{(j)} - \theta^{(j-1)} \right\|_2 \leq \eta M R_0 \sum_{j=1}^{t} B^{j-1} < \eta^* \frac{M B^{T_1} R_0}{B - 1} \tag{6}$$

*Proof.* To prove Lemma 2, we need the following result:

**Lemma 3** (Lemma 13 in Zhai et al. (2022))**.** *Under Assumption 1, there is a constant $M > 0$ such that for any $C_0 > 0$ and any $\delta > 0$, there exists a $\tilde{D}$ such that: If $\tilde{d} \geq \tilde{D}$, then with probability at least $(1 - \delta)$ over random initialization, for any $\boldsymbol{x}$ such that $\|\boldsymbol{x}\|_2 \leq 1$,*

$$\left\| \nabla_\theta f(\boldsymbol{x}; \theta) - \nabla_\theta f(\boldsymbol{x}; \tilde{\theta}) \right\|_2 \leq \frac{M}{\sqrt[4]{\tilde{d}}} \|\theta - \tilde{\theta}\|_2 \tag{7a}$$

$$\left\| \nabla_\theta f(\boldsymbol{x}; \theta) \right\|_2 \leq M \tag{7b}$$

$$\|J(\theta) - J(\tilde{\theta})\|_F \leq \frac{M}{\sqrt[4]{\tilde{d}}} \|\theta - \tilde{\theta}\|_2, \quad \forall \theta, \tilde{\theta} \in B\left(\theta^{(0)}, C_0\right) \tag{7c}$$

$$\|J(\theta)\|_F \leq M \tag{7d}$$

*for $\forall \theta, \tilde{\theta} \in B\left(\theta^{(0)}, C_0\right)$, where $B\left(\theta^{(0)}, R\right) = \left\{ \theta : \left\| \theta - \theta^{(0)} \right\|_2 < R \right\}$.*

Note that for any $\boldsymbol{x}, f^{(0)}(\boldsymbol{x}) = 0$. Thus, for any $\delta > 0$, there exists a constant $R_0$ such that with probability at least $(1 - \delta/3)$ over random initialization,

$$\left\| u\left(\theta^{(0)}\right) \right\|_2 < R_0 \tag{8}$$

By the NTK result:

**Lemma 4.** *If $\sigma$ is Lipschitz and $d_l \to \infty$ for $l = 1, \cdots, L$ sequentially, then $K^{(0)}\left(\boldsymbol{x}, \boldsymbol{x}'\right)$ converges in probability to a non-degenerate deterministic limiting kernel $K\left(\boldsymbol{x}, \boldsymbol{x}'\right)$.*

Let $M$ be the constant in Lemma 3. Let $B = 1 + \eta^* M^2$, and $C_0 = \eta^* \frac{MB^{T_1}R_0}{B-1}$. By Lemma 3, there exists $D_1 > 0$ such that with probability at least $(1 - \delta/3)$, for any $\tilde{d} \geq D_1$, 7 is true for all $\theta, \tilde{\theta} \in B\left(\theta^{(0)}, C_0\right)$.

By union bound, with probability at least $1 - \delta$, 7, 8 hold. We prove 5 and 6 by induction.

$$
\begin{aligned}
\left\|\theta^{(t+1)} - \theta^{(t)}\right\|_2 &\leq \eta \left\|J\left(\theta^{(t)}\right) W^{(t)}\right\|_2 \left\|u\left(\theta^{(t)}\right)\right\|_2 \\
&\leq \eta \left\|J\left(\theta^{(t)}\right) W^{(t)}\right\|_F \left\|u\left(\theta^{(t)}\right)\right\|_2 \\
&\leq \eta \left\|J\left(\theta^{(t)}\right)\right\|_F \left\|u\left(\theta^{(t)}\right)\right\|_2 \\
&\leq \eta M B^t R_0
\end{aligned}
$$

$$
\begin{aligned}
\left\|u\left(\theta^{(t+1)}\right)\right\|_2 &= \left\|u\left(\theta^{(t+1)}\right) - u\left(\theta^{(t)}\right) + u\left(\theta^{(t)}\right)\right\|_2 \\
&= \left\|J\left(\tilde{\theta}^{(t)}\right)^\top \left(\theta^{(t+1)} - \theta^{(t)}\right) + u\left(\theta^{(t)}\right)\right\|_2 \\
&= \left\|-\eta J\left(\tilde{\theta}^{(t)}\right)^\top J\left(\theta^{(t)}\right) W^{(t)} u\left(\theta^{(t)}\right) + u\left(\theta^{(t)}\right)\right\|_2 \\
&\leq \left\|\boldsymbol{I} - \eta J\left(\tilde{\theta}^{(t)}\right)^\top J\left(\theta^{(t)}\right) W^{(t)}\right\|_2 \left\|u\left(\theta^{(t)}\right)\right\|_2 \\
&\leq \left(1 + \left\|\eta J\left(\tilde{\theta}^{(t)}\right)^\top J\left(\theta^{(t)}\right) W^{(t)}\right\|_2\right) \left\|u\left(\theta^{(t)}\right)\right\|_2 \\
&\leq \left(1 + \eta \left\|J\left(\tilde{\theta}^{(t)}\right)\right\|_F \left\|J\left(\theta^{(t)}\right)\right\|_F\right) \left\|u\left(\theta^{(t)}\right)\right\|_2 \\
&\leq \left(1 + \eta M^2\right) \left\|u\left(\theta^{(t)}\right)\right\|_2 \leq B^{t+1} R_0
\end{aligned}
$$

$\qquad\qquad\qquad\qquad\qquad\qquad\qquad\qquad\qquad\qquad\qquad\qquad\qquad\qquad\quad\square$

Now start to prove Theorem 2.

*Proof.* Denote $\Delta_t = u_{\text{lin}}\left(\theta^{(t)}\right) - u\left(\theta^{(t)}\right)$. Then

$$
\begin{cases}
u_{\text{lin}}\left(\theta^{(t+1)}\right) - u_{\text{lin}}\left(\theta^{(t)}\right) = -\eta J\left(\theta^{(0)}\right)^\top J\left(\theta^{(0)}\right) W^{(t)} u_{\text{lin}}\left(\theta^{(t)}\right) \\
u\left(\theta^{(t+1)}\right) - u\left(\theta^{(t)}\right) = -\eta J\left(\tilde{\theta}^{(t)}\right)^\top J\left(\theta^{(t)}\right) W^{(t)} u\left(\theta^{(t)}\right)
\end{cases}
\tag{9}
$$

Thus

$$
\begin{aligned}
\Delta_{t+1} - \Delta_t =\,& \eta \left[J\left(\tilde{\theta}^{(t)}\right)^\top J\left(\theta^{(t)}\right) - J\left(\theta^{(0)}\right)^\top J\left(\theta^{(0)}\right)\right] W^{(t)} u\left(\theta^{(t)}\right) \\
& - \eta J\left(\theta^{(0)}\right)^\top J\left(\theta^{(0)}\right) W^{(t)} \Delta_t
\end{aligned}
$$

By Lemma 3,

$$
\begin{aligned}
& \left\|J\left(\tilde{\theta}^{(t)}\right)^\top J\left(\theta^{(t)}\right) - J\left(\theta^{(0)}\right)^\top J\left(\theta^{(0)}\right)\right\|_F \\
& \leq \left\|\left(J\left(\tilde{\theta}^{(t)}\right) - J\left(\theta^{(0)}\right)\right)^\top J\left(\theta^{(t)}\right)\right\|_F + \left\|J\left(\theta^{(0)}\right)^\top \left(J\left(\theta^{(t)}\right) - J\left(\theta^{(0)}\right)\right)\right\|_F \\
& \leq 2M^2 C_0 \tilde{d}^{-1/4}
\end{aligned}
$$

Then

$$\|\Delta_{t+1}\|_2$$

$$\leq \left\| \left[ \boldsymbol{I} - \eta J\left(\theta^{(0)}\right)^\top J\left(\theta^{(0)}\right) W^{(t)} \right] \Delta_t \right\|_2 + \left\| \eta \left[ J\left(\tilde{\theta}^{(t)}\right)^\top J\left(\theta^{(t)}\right) - J\left(\theta^{(0)}\right)^\top J\left(\theta^{(0)}\right) \right] W^{(t)} u\left(\theta^{(t)}\right) \right\|_2$$

$$\leq \left\| \boldsymbol{I} - \eta J\left(\theta^{(0)}\right)^\top J\left(\theta^{(0)}\right) W^{(t)} \right\|_F \|\Delta_t\|_2 + \eta \left\| J\left(\tilde{\theta}^{(t)}\right)^\top J\left(\theta^{(t)}\right) - J\left(\theta^{(0)}\right)^\top J\left(\theta^{(0)}\right) \right\|_F \left\| u\left(\theta^{(t)}\right) \right\|_2$$

$$\leq \left(1 + \eta M^2\right) \|\Delta_t\|_2 + 2\eta M^2 C_0 B^t R_0 \tilde{d}^{-1/4}$$

$$\leq B \|\Delta_t\|_2 + 2\eta M^2 C_0 B^t R_0 \tilde{d}^{-1/4}$$

Therefore,

$$B^{-(t+1)} \|\Delta_{t+1}\|_2 \leq B^{-t} \|\Delta_t\|_2 + 2\eta M^2 C_0 B^{-1} R_0 \tilde{d}^{-1/4}.$$

Since $\Delta_0 = 0$, we have

$$\|\Delta_t\|_2 \leq 2t\eta M^2 C_0 B^{t-1} R_0 \tilde{d}^{-1/4}.$$

$\square$

Proof of Theorem 1:

*Proof.* We first calculate $\nabla_w \sum_{j=1}^m l_j^v(\theta_t - \eta\nabla\sum_{i=1}^n w_t^i l_i(\theta_t))$.

By a direct computation, we notice

$$\frac{\partial}{\partial w^k} l_j^v(\theta_t - \eta\nabla\sum_{i=1}^n w_t^i l_i(\theta_t))$$

$$= (f_j^v(\theta_t - \eta\nabla\sum_{i=1}^n w_t^i l_i(\theta_t)) - y_j^v) \frac{\partial f_j^v(\theta_t - \eta\nabla\sum_{i=1}^n w_t^i l_i(\theta_t))}{\partial w^k}$$

where

$$\frac{d f_j^v(\theta_t - \eta\nabla_\theta w_t^T l(\theta_t))}{dw^k} = \langle \nabla_\theta f_j^v(\theta_t - \eta\nabla_\theta w_t^T l(\theta_t)), \frac{d}{dw^k}(\theta_t - \eta\nabla_\theta w_t^T l(\theta_t)) \rangle$$

$$= \langle \nabla_\theta f_j^v(\theta_t - \eta\nabla_\theta w_t^T l(\theta_t)), -\eta\nabla_\theta l_k(\theta_t) \rangle$$

$$= \langle \nabla_\theta f_j^v(\theta_t - \eta\nabla_\theta w_t^T l(\theta_t)), -\eta(f_k(\theta_t) - y_k)\nabla f_k(\theta_t) \rangle$$

Therefore,

$$\frac{\partial}{\partial w^k} l_j^v(\theta_t - \eta\nabla\sum_{i=1}^n w_t^i l_i(\theta_t))$$

$$= -\eta u_j^v(\theta_t - \eta\nabla\sum_{i=1}^n w_t^i l_i(\theta_t)) u_k(\theta_t) \langle \nabla_\theta f_j^v(\theta_t - \eta\nabla_\theta w_t^T l(\theta_t)), \nabla_\theta f_k(\theta_t) \rangle$$

and consequently

$$\sum_{j=1}^m \frac{\partial}{\partial w^k} l_j^v(\theta_t - \eta\nabla\sum_{i=1}^n w_t^i l_i(\theta_t))$$

$$= -\eta\langle \sum_{j=1}^m u_j^v(\theta_t - \eta\nabla\sum_{i=1}^n w_t^i l_i(\theta_t))\nabla_\theta f_j^v(\theta_t - \eta\nabla_\theta w_t^T l(\theta_t)), u_k(\theta_t)\nabla_\theta f_k(\theta_t) \rangle.$$

So the weight gradient can be computed as

$$\nabla_w \sum_{j=1}^{m} l_j^v(\theta_t - \eta\nabla\sum_{i=1}^{n} w_t^i l_i(\theta_t))$$

$$= -\eta\Big(J(\theta_t)U(\theta_t)\Big)^T\Big(\sum_{j=1}^{m} u_j^v(\theta_t - \eta\nabla\sum_{i=1}^{n} w_t^i l_i(\theta_t))\nabla_\theta f_j^v(\theta_t - \eta\nabla_\theta w_t^T l(\theta_t))\Big)$$

$$= -\eta\Big(J(\theta_t)U(\theta_t)\Big)^T\Big(J^v(\theta_t - \eta\nabla_\theta w_t^T l(\theta_t))u^v(\theta_t - \eta\nabla_\theta w_t^T l(\theta_t))\Big)$$

$$= -\eta U(\theta_t)J(\theta_t)^T J^v(\theta_t - \eta\nabla_\theta w_t^T l(\theta_t))u^v(\theta_t - \eta\nabla_\theta w_t^T l(\theta_t))$$

$$= -\eta U(\theta_t)J(\theta_t)^T\Big(J^v(\theta_t) + J^v(\theta_t - \eta\nabla_\theta w_t^T l(\theta_t)) - J^v(\theta_t)\Big)\Big(u^v(\theta_t) + u^v(\theta_t - \eta\nabla_\theta w_t^T l(\theta_t)) - u^v(\theta_t)\Big)$$

$$\approx -\eta U(\theta_t)J(\theta_t)^T J^v(\theta_t)u^v(\theta_t) + O(\eta^2 error),$$

where $U(\theta_t)$ denote the diagonal matrix from $u(\theta_t)$. To estimate the approximation error in the last step, we observe

$$\|J^v(\theta_t - \eta\nabla_\theta w_t^T l(\theta_t)) - J^v(\theta_t)\|_F \leq \frac{M}{\sqrt[4]{\tilde{d}}}\|\eta\nabla_\theta w_t^T l(\theta_t)\|_2 \leq \frac{M^2}{\sqrt[4]{\tilde{d}}}\eta B^t R_0$$

and

$$\|u^v(\theta_t - \eta\nabla_\theta w_t^T l(\theta_t)) - u^v(\theta_t)\|_2 = \|\eta J^v(\tilde{\theta})^T\nabla_\theta w_t^T l(\theta_t)\|_2 \leq \eta M^2 B^t R_0.$$

As a consequence,

$$\|\nabla_w \sum_{j=1}^{m} l_j^v(\theta_t - \eta\nabla\sum_{i=1}^{n} w_t^i l_i(\theta_t)) + \eta U(\theta_t)J(\theta_t)^T J^v(\theta_t)u^v(\theta_t)\|_2$$

$$= \|\eta U(\theta_t)J(\theta_t)^T\Big(J^v(\theta_t) + J^v(\theta_t - \eta\nabla_\theta w_t^T l(\theta_t)) - J^v(\theta_t)\Big)\Big(u^v(\theta_t) + u^v(\theta_t - \eta\nabla_\theta w_t^T l(\theta_t)) - u^v(\theta_t)\Big)\|_2$$

$$= \|\eta U(\theta_t)J(\theta_t)^T\Big(J^v(\theta_t - \eta\nabla_\theta w_t^T l(\theta_t)) - J^v(\theta_t)\Big)u^v(\theta_t)\|_2$$

$$+ \|\eta U(\theta_t)J(\theta_t)^T J^v(\theta_t)\Big(u^v(\theta_t - \eta\nabla_\theta w_t^T l(\theta_t)) - u^v(\theta_t)\Big)\|_2$$

$$+ \|\eta U(\theta_t)J(\theta_t)^T\Big(J^v(\theta_t - \eta\nabla_\theta w_t^T l(\theta_t)) - J^v(\theta_t)\Big)\Big(u^v(\theta_t - \eta\nabla_\theta w_t^T l(\theta_t)) - u^v(\theta_t)\Big)\|_2$$

$$\lesssim \eta B^t R_0 M\Big(\frac{M^2}{\sqrt[4]{\tilde{d}}}\eta B^{2t} R_0^2 + \eta M^2 B^t R_0 + \frac{M^4}{\sqrt[4]{\tilde{d}}}\eta^2 B^{3t} R_0^3\Big)$$

$$\lesssim \eta^2 B^{2t} R_0^2 M^3.$$

Together with

$$\|\boldsymbol{K}^{(t)} - \boldsymbol{K}^{(0)}\|_2$$

$$= \Big\|J\left(\theta^{(t)}\right)^\top J\left(\theta^{(t)}\right) - J\left(\theta^{(0)}\right)^\top J\left(\theta^{(0)}\right)\Big\|_2$$

$$\leq \Big\|\left(J\left(\theta^{(t)}\right) - J\left(\theta^{(0)}\right)\right)^\top J\left(\theta^{(t)}\right)\Big\|_2 + \Big\|J\left(\theta^{(0)}\right)^\top\left(J\left(\theta^{(t)}\right) - J\left(\theta^{(0)}\right)\right)\Big\|_2$$

$$\leq 2M^2 C_0 \tilde{d}^{-1/4},$$

we get

$$\|w_{t+1} - \Big(w_t + \eta\alpha U(\theta_t)K^{(0)}(X, X_{clean})u^v(\theta_t)\Big)\|_2$$

$$\lesssim \eta^2\alpha B^{2t} R_0^2 M^3 + \eta\alpha\|U(\theta_t)K^{(t)}(X, X_{clean})u^v(\theta_t) - U(\theta_t)K^{(0)}(X, X_{clean})u^v(\theta_t)\|_2$$

$$\lesssim \alpha\eta^2 B^{2t} R_0^2 M^3 + \alpha\eta B^{2t} R_0^2 M^2 C_0 \tilde{d}^{-1/4}.$$

Now we estimate the difference of replacing $\boldsymbol{u}$ by $\boldsymbol{y}$:

$$\|U(\theta_t)K^{(0)}(X_{tr}, X_{val})u^v(\theta_t) - U_0 K^{(0)}(X_{tr}, X_{val})u_0^v\|_\infty$$

$$= \|(U_t - U_0)K^{(0)}(X_{tr}, X_{val})u_0^v\|_\infty + \|U_0 K^{(0)}(X_{tr}, X_{val})(u_t^v - u_0^v)\|_\infty$$

$$+ \|(U_t - U_0)K^{(0)}(X_{tr}, X_{val})(u_t^v - u_0^v)\|_\infty.$$

Since $\boldsymbol{u}_t - \boldsymbol{u}_0$ can be bounded via:

$$
\begin{aligned}
\|\boldsymbol{u}_t - \boldsymbol{u}_0\|_\infty &= \|\boldsymbol{u}_t - \boldsymbol{u}_{lin}^{(t)} + \boldsymbol{u}_{lin}^{(t)} - \boldsymbol{u}_0\|_\infty \\
&\leq \|\boldsymbol{u}_t - \boldsymbol{u}_{lin}^{(t)}\|_\infty + \|\boldsymbol{u}_{lin}^{(t)} - \boldsymbol{u}_0\|_\infty \\
&\leq C\eta t B^{t-1}\tilde{d}^{-1/4} + \|\boldsymbol{u}_{lin}^{(t)} - \boldsymbol{u}_0\|_2 \\
&= C\eta t B^{t-1}\tilde{d}^{-1/4} + \|f_{\text{lin}}^{(t)}(X) - f_{\text{lin}}^{(0)}(X)\|_2 \\
&= C\eta t B^{t-1}\tilde{d}^{-1/4} + \|\nabla_\theta f^{(0)}(X)^T(\theta^{(t)} - \theta^{(0)})\|_2 \\
&\leq C\eta t B^{t-1}\tilde{d}^{-1/4} + \sqrt{\lambda_{max}}\|\theta^{(t)} - \theta^{(0)}\|_2 \\
&\leq C\eta t B^{t-1}\tilde{d}^{-1/4} + \sqrt{\lambda_{max}}\eta C_0.
\end{aligned}
$$

Combine the previous error bounds, we get

$$
\begin{aligned}
&\|w_{t+1} - (w_t + \eta\alpha U_0 K^{(0)}(X_{tr}, X_{val})u_0^v)\|_\infty \\
&\leq \|w_{t+1} - (w_t + \eta\alpha U(\theta_t)K^{(0)}(X_{tr}, X_{val})u^v(\theta_t))\|_\infty \\
&\quad + \alpha\eta\|U(\theta_t)K^{(0)}(X_{tr}, X_{val})u^v(\theta_t) - U_0 K^{(0)}(X_{tr}, X_{val})u_0^v\|_\infty \\
&\lesssim \alpha\eta^2 B^{2t}R_0^2 M^3 + \alpha\eta B^{2t}R_0^2 M^2 C_0\tilde{d}^{-1/4} + C\alpha\eta^2\lambda_{max}\sqrt{m}tB^{t-1}\tilde{d}^{-1/4} + \alpha\eta^2\lambda_{max}^{3/2}C_0 \\
&\lesssim \alpha\eta^2 B^{2t} + \alpha\eta B^{2t}C_0\tilde{d}^{-1/4} + \alpha\eta^2\lambda_{max}\sqrt{m}tB^{t-1}\tilde{d}^{-1/4} + \alpha\eta^2\lambda_{max}^{3/2}C_0 \\
&\lesssim \alpha\eta^2\lambda_{max}^{3/2}B^{2T_1} + \alpha\eta^2\lambda_{max}\sqrt{m}T_1 B^{T_1}\tilde{d}^{-1/4} + \alpha\eta B^{3T_1}\tilde{d}^{-1/4}.
\end{aligned}
$$

Thus, together with Assumption 3, under a suitable choice of $\eta^*$ and $\tilde{D}$, for any $\eta < \eta^*$ and $\tilde{d} > \tilde{D}$, we have $w_{T_1}^{(i)} = 0$ for noisy samples and $w_{T_1}^{(i)} = 1$ for clean samples at $T_1 = 1 + (m\alpha\eta\gamma)^{-1}$. $\qquad\square$

## B.2 Filtering Phase

We analyze the algorithm for $T_1 \leq t \leq T_2$. Note that since the error bound B.1 could increase as $t$ increases, and in order for the training error to converge, it is necessary to have a sufficiently large $t$, we need a different way to control the weights $\boldsymbol{w}$ while obtaining convergence of the training loss. For this section, we prove

$$
w_t^i = \begin{cases} 1, & y_i = y_i^*, \\ 0, & y_i \neq y_i^*. \end{cases}
$$

by induction. We denote $W^*$ as the diagonal matrix formed by $w^i$. Specifically, we show

**Lemma 5.** *There exist constants $M > 0$ such that for any $\delta > 0$, there exist $R_0 > 0, \tilde{D} > 0$ and $B > 1$ such that for any $\tilde{d} \geq \tilde{D}$, the following hold with probability at least $(1 - \delta)$ over random initialization when applying gradient descent with learning rate $\eta$ : For all $T_1 \leq t \leq T_2$, we have*

$$
\left\|W^* u\left(\theta^{(t)}\right)\right\|_2 \leq \left(1 - \frac{\eta\lambda_{\min}}{3}\right)^{t-T_1} B^{T_1} R_0 \tag{10}
$$

$$
\sum_{j=T_1+1}^{t}\left\|\theta^{(j)} - \theta^{(j-1)}\right\|_2 \leq \eta M B^{T_1} R_0 \sum_{j=T_1+1}^{t}\left(1 - \frac{\eta\lambda_{\min}}{3}\right)^{j-T_1} \tag{11}
$$

$$
< \frac{3MB^{T_1}R_0}{\lambda_{\min}} \tag{12}
$$

$$
w_t^i = \begin{cases} 1, & y_i = y_i^*, \\ 0, & y_i \neq y_i^*. \end{cases} \tag{13}
$$

*Proof.* There exists $D_2 \geq 0$ such that for any $\tilde{d} \geq D_2$, with probability at least $(1 - \delta/3)$,

$$
\left\|\boldsymbol{K} - \boldsymbol{K}^{(0)}\right\|_F \leq \frac{\lambda_{\min}}{3}. \tag{14}
$$

We prove this lemma by induction. We first check the conclusions when $t = T_1$. By 5, $\left\| W^* u \left( \theta^{(t)} \right) \right\|_2 \leq \| u \left( \theta^{(t)} \right) \| \leq B^{T_1} R_0$. 11 holds trivially. 13 holds due to the choice of $T_1$.

Then for $t + 1$,

$$
\begin{aligned}
\left\| \theta^{(t+1)} - \theta^{(t)} \right\|_2 &\leq \eta \left\| J \left( \theta^{(t)} \right) \right\|_2 \left\| W^* u \left( \theta^{(t)} \right) \right\|_2 \\
&\leq \eta \left\| J \left( \theta^{(t)} \right) \right\|_F \left\| W^* u \left( \theta^{(t)} \right) \right\|_2 \\
&\leq M \eta \left( 1 - \frac{\eta \lambda_{\min}}{3} \right)^{t - T_1} B^{T_1} R_0
\end{aligned}
\tag{15}
$$

Consequently,

$$
\begin{aligned}
\left\| W^* u \left( \theta^{(t+1)} \right) \right\|_2 &= \left\| W^* u \left( \theta^{(t+1)} \right) - W^* u \left( \theta^{(t)} \right) + W^* u \left( \theta^{(t)} \right) \right\|_2 \\
&= \left\| W^* J \left( \tilde{\theta}^{(t)} \right)^\top \left( \theta^{(t+1)} - \theta^{(t)} \right) + W^* u \left( \theta^{(t)} \right) \right\|_2 \\
&= \left\| -\eta W^* J \left( \tilde{\theta}^{(t)} \right)^\top J \left( \theta^{(t)} \right) W^* u \left( \theta^{(t)} \right) + W^* u \left( \theta^{(t)} \right) \right\|_2 \\
&\leq \left\| \boldsymbol{I} - \eta W^* J \left( \tilde{\theta}^{(t)} \right)^\top J \left( \theta^{(t)} \right) W^* \right\|_{2, W^*} \left\| W^* u \left( \theta^{(t)} \right) \right\|_2 \\
&\leq \left\| \boldsymbol{I} - \eta W^* J \left( \tilde{\theta}^{(t)} \right)^\top J \left( \theta^{(t)} \right) W^* \right\|_{2, W^*} \left( 1 - \frac{\eta \lambda_{\min}}{3} \right)^{t - T_1} B^{T_1} R_0,
\end{aligned}
$$

where $\| \cdot \|_{2, W^*}$ indicates the 2-norm restricted on the image space of $W^*$.

Since

$$
\begin{aligned}
&\left\| \boldsymbol{I} - \eta W^* J \left( \tilde{\theta}^{(t)} \right)^\top J \left( \theta^{(t)} \right) W^* \right\|_{2, W^*} \\
&\leq \| \boldsymbol{I} - \eta W^* K W^* \|_{2, W^*} + \eta \left\| W^* \left( K - K^{(0)} \right) W^* \right\|_{2, W^*} \\
&\quad + \eta \left\| W^* \left( J \left( \theta^{(0)} \right)^\top J \left( \theta^{(0)} \right) - J \left( \tilde{\theta}^{(t)} \right)^\top J \left( \theta^{(t)} \right) \right) W^* \right\|_{2, W^*} \\
&\leq 1 - \eta \lambda_{\min} + \eta \left\| K - K^{(0)} \right\|_F + \eta \left\| J \left( \theta^{(0)} \right)^\top J \left( \theta^{(0)} \right) - J \left( \tilde{\theta}^{(t)} \right)^\top J \left( \theta^{(t)} \right) \right\|_F \\
&\leq 1 - \eta \lambda_{\min} + \frac{\eta \lambda_{\min}}{3} + \frac{\eta M^2}{\sqrt[4]{\tilde{d}}} \left( \left\| \theta^{(t)} - \theta^{(0)} \right\|_2 + \left\| \tilde{\theta}^{(t)} - \theta^{(0)} \right\|_2 \right) \\
&\leq 1 - \frac{\eta \lambda_{\min}}{3}
\end{aligned}
$$

for $\tilde{d} \geq \max \left\{ D_1, D_2, \left( \frac{6 M^2 C_0}{\lambda_{\min}} \right)^4 \right\}$.

Therefore,

$$
\left\| W^* u \left( \theta^{(t+1)} \right) \right\|_2 \leq \left( 1 - \frac{\eta \lambda^{\min}}{3} \right)^{t+1-T_1} B^{T_1} R_0.
$$

Now we can improve our linear approximation error to be uniform on $t$.

Again, by writing

$$\nabla_w \sum_{j=1}^{m} l_j^v(\theta_t - \eta \nabla \sum_{i=1}^{n} w_t^i l_i(\theta_t))$$

$$= -\eta U(\theta_t) J(\theta_t)^T \Big( J^v(\theta_t) + J^v(\theta_t - \eta \nabla_\theta w_t^T l(\theta_t)) - J^v(\theta_t) \Big) \Big( u^v(\theta_t) + u^v(\theta_t - \eta \nabla_\theta w_t^T l(\theta_t)) - u^v(\theta_t) \Big)$$

$$= -\eta U(\theta_t) \Big[ J(\theta_t)^T J^v(\theta_t) u^v(\theta_t) - J(\theta_t)^T \big( J^v(\theta_t - \eta \nabla_\theta w_t^T l(\theta_t)) - J^v(\theta_t) \big) u^v(\theta_t)$$

$$- J(\theta_t)^T J^v(\theta_t) \big( u^v(\theta_t - \eta \nabla_\theta w_t^T l(\theta_t)) - u^v(\theta_t) \big)$$

$$+ J(\theta_t)^T \big( J^v(\theta_t - \eta \nabla_\theta w_t^T l(\theta_t)) - J^v(\theta_t) \big) \big( u^v(\theta_t - \eta \nabla_\theta w_t^T l(\theta_t)) - u^v(\theta_t) \big) \Big]$$

and with the bounds

$$\|J^v(\theta_t - \eta \nabla_\theta w_t^T l(\theta_t)) - J^v(\theta_t)\|_F \le \frac{M}{\sqrt[4]{\tilde{d}}} \|\eta \nabla_\theta w_t^T l(\theta_t)\|_2 \lesssim \frac{M^2}{\sqrt[4]{\tilde{d}}} \eta,$$

$$\|u^v(\theta_t - \eta \nabla_\theta w_t^T l(\theta_t)) - u^v(\theta_t)\|_2 = \|\eta J^v(\tilde{\theta})^T \nabla_\theta w_t^T l(\theta_t)\|_2 \lesssim \eta M^2,$$

we have

$$\|J(\theta_0)^T J^v(\theta_0) u^v(\theta_t) - \Big[ J(\theta_t)^T J^v(\theta_t) u^v(\theta_t) - J(\theta_t)^T \big( J^v(\theta_t - \eta \nabla_\theta w_t^T l(\theta_t)) - J^v(\theta_t) \big) u^v(\theta_t)$$

$$- J(\theta_t)^T J^v(\theta_t) \big( u^v(\theta_t - \eta \nabla_\theta w_t^T l(\theta_t)) - u^v(\theta_t) \big)$$

$$+ J(\theta_t)^T \big( J^v(\theta_t - \eta \nabla_\theta w_t^T l(\theta_t)) - J^v(\theta_t) \big) \big( u^v(\theta_t - \eta \nabla_\theta w_t^T l(\theta_t)) - u^v(\theta_t) \big) \Big] \|_\infty$$

$$\lesssim \|J(\theta_0)^T J^v(\theta_0) u^v(\theta_t) - J(\theta_t)^T J^v(\theta_t) u^v(\theta_t)\|_\infty + \eta M^3 \tilde{d}^{-1/4} + \eta M^4$$

$$\lesssim \tilde{d}^{-1/4} + \eta.$$

Finally, we observe from $u_t^i = u_{t-1}^i - \eta \sum_{j=1}^n K_{ij} w_t^j u_{t-1}^j + O(\tilde{C} \tilde{d}^{-1/4})$ that for noisy sample indices $i$, (1) $w_t^j = 0$ for noisy sample indices $j$; (2) $sgn(K_{ij} u_t^j) = -sgn(u_t^i)$ for clean indices $j$. As a result, $u_t^i$ keeps the sign while obtaining larger magnitude. Similarly, for clean samples $u_{t+1}^i \to B(0, r)$ monotonically after $T_1$ for some $r = O(\eta + \tilde{d}^{-1/4})$. Therefore, the noisy sample filtering keeps until some $u_i^v \lesssim \eta + \tilde{d}^{-1/4}$. With the following additional Assumption 5, the noisy sample filtering phase can be extend until $\|\boldsymbol{u}^v\|_\infty \lesssim \eta + \tilde{d}^{-1/4}$.

**Assumption 5.** *Denote the decomposition of $\boldsymbol{u}^v$ as*

$$u_i^v(t) = \sum_{j=1}^{n} s_{ij} e^{-\eta \lambda_j t}.$$

*Let $j^* := \arg\min\{\lambda_j > 0 : s_{ij} \neq 0\}$ and define $s_i^* := s_{ij^*}, \lambda_i^* := \lambda_{j^*}$. Let $t_i$ be the first time such that $|u_i^v(t_i)| \lesssim \eta + \tilde{d}^{-1/4}$. Assume for $\forall t \ge t_i$,*

$$\mathrm{sgn}\big(u_i^v(t)\big) = \mathrm{sgn}\big(s_i^*\big).$$

The noise sample filtering fails for $u^v(\theta_t)$ when the weight update signal is overwhelmed by the approximation error bound, i.e.

$$\gamma \sum_k |u_k^v| \lesssim \tilde{d}^{-1/4} + \eta.$$

$\square$

## C  PROOF OF PROPOSITION 1

*Proof.* **Upper bound for $C$:** Let $s := K_m \mathbf{1}_m$. Since $\mathbf{1}_m^\top u^v = 0$,

$$C = -\frac{1}{m} s^\top u^v = -\frac{1}{m} (s - c_m \mathbf{1}_m)^\top u^v.$$

By the Cauchy–Schwarz inequality and equation 4,

$$|C| \;\leq\; \frac{1}{m}\,\|s - c_m \mathbf{1}_m\|_2\,\|u^v\|_2 \;=\; \frac{1}{m}\cdot O_p\big(m^{1-\varepsilon}\big)\cdot\sqrt{m} \;=\; O_p\big(m^{\frac{1}{2}-\varepsilon}\big).$$

**Lower-order growth for $S_i$:** Assume $x_i$ is independent of $X_{\text{clean}}$. Define the row-wise mean $\mu_K(x_i) := \mathbb{E}_{X'\sim\mathcal{D}}[K(x_i, X')]$ and the centered row vector $k_i^\circ := K(x_i, X_{\text{clean}}) - \mu_K(x_i)\,\mathbf{1}_m^\top \in \mathbb{R}^m$. Because $\mathbf{1}_m^\top u^v = 0$,

$$S_i \;=\; K(x_i, X_{\text{clean}})\,u^v \;=\; \big(k_i^\circ + \mu_K(x_i)\mathbf{1}_m^\top\big)u^v \;=\; (k_i^\circ)^\top u^v.$$

Let $Z_j := K(x_i, X_j^{\text{clean}}) - \mu_K(x_i)$ for $j = 1, \ldots, m$. Conditional on $x_i$, the variables $Z_1, \ldots, Z_m$ are i.i.d., satisfy $\mathbb{E}[Z_j \mid x_i] = 0$ and $|Z_j| \leq 2K_{\max}$, and $\mathrm{Var}(Z_j \mid x_i) \geq \sigma^2$ by the non-degeneracy assumption. Since $S_i = \sum_{j=1}^m u_j^v Z_j$ with $|u_j^v| = 1$,

$$\mathbb{E}\big[S_i^2 \mid x_i\big] = \sum_{j=1}^m (u_j^v)^2\,\mathrm{Var}(Z_j \mid x_i) \;\geq\; m\,\sigma^2.$$

Moreover, boundedness yields a finite fourth moment: there exists a constant $C_4 > 0$ (depending only on $K_{\max}$) such that $\mathbb{E}[S_i^4 \mid x_i] \leq C_4\, m^2 K_{\max}^4$. Applying the Paley–Zygmund inequality to $S_i^2$ conditional on $x_i$ with parameter $\gamma = \frac{1}{2}$ gives

$$\mathbb{P}\bigg(|S_i| \;\geq\; \tfrac{1}{2}\sqrt{\mathbb{E}[S_i^2 \mid x_i]}\;\bigg|\;x_i\bigg) \;\geq\; (1 - \tfrac{1}{2})^2\,\frac{\mathbb{E}[S_i^2 \mid x_i]^2}{\mathbb{E}[S_i^4 \mid x_i]} \;\geq\; \frac{\tfrac{1}{4}\,m^2\sigma^4}{C_4\,m^2 K_{\max}^4} \;=:\; c_1 \in (0, 1),$$

hence $\mathbb{P}(|S_i| \geq (\sigma/2)\sqrt{m}) \geq c_1$.

For a high-probability lower bound, set $V_i^2 := \mathrm{Var}(S_i \mid x_i) = \sum_{j=1}^m (u_j^v)^2\,\mathrm{Var}(Z_j \mid x_i) \geq m\sigma^2$. By Berry–Esseen (bounded third moments since $|Z_j| \leq 2K_{\max}$),

$$\sup_{x \in \mathbb{R}}\bigg|\mathbb{P}\bigg(\frac{S_i}{V_i} \leq x\;\bigg|\;x_i\bigg) - \Phi(x)\bigg| \;\leq\; \frac{C_{\mathrm{BE}}\sum_{j=1}^m \mathbb{E}\big[|u_j^v Z_j|^3 \mid x_i\big]}{V_i^3} \;\leq\; \frac{C'}{\sqrt{m}},$$

for a constant $C'$ depending only on $K_{\max}$ and $\sigma$. Let $\tau_m := m^{-\varepsilon/2}$. Since $V_i \geq \sigma\sqrt{m}$,

$$\mathbb{P}\big(|S_i| \leq \tau_m\sqrt{m}\;\big|\;x_i\big) \;\leq\; 2\Phi\bigg(\frac{\tau_m\sqrt{m}}{V_i}\bigg) - 1 + \frac{C'}{\sqrt{m}} \;\leq\; \frac{2}{\sqrt{2\pi}}\frac{\tau_m\sqrt{m}}{V_i} + \frac{C'}{\sqrt{m}} \;\leq\; \frac{2}{\sqrt{2\pi}\,\sigma}\,m^{-\varepsilon/2} + \frac{C'}{\sqrt{m}}.$$

Taking expectation over $x_i$ yields

$$\mathbb{P}\bigg(|S_i| \leq m^{\frac{1}{2}-\frac{\varepsilon}{2}}\bigg) \;\leq\; C''\,m^{-\varepsilon/2} + C'/\sqrt{m} \;\xrightarrow[m\to\infty]{}\; 0,$$

for a constant $C'' > 0$. Consequently,

$$|S_i| \;\geq\; m^{\frac{1}{2}-\frac{\varepsilon}{2}} \quad \text{with probability } 1 - o(1),$$

and in particular $|S_i| = \Omega_p(\sqrt{m})$.

*Combining with the bound for $C$:* Since $|C| = O_p\big(m^{\frac{1}{2}-\varepsilon}\big)$ and $|S_i| \geq m^{\frac{1}{2}-\frac{\varepsilon}{2}}$ with probability $1 - o(1)$, it follows that

$$\frac{|C|}{|S_i|} \;=\; O_p\bigg(\frac{m^{\frac{1}{2}-\varepsilon}}{m^{\frac{1}{2}-\frac{\varepsilon}{2}}}\bigg) \;=\; O_p\big(m^{-\frac{\varepsilon}{2}}\big) \;\xrightarrow{p}\; 0.$$

$\square$

Table 3: Test Accuracy on CIFAR-10N/100N.

| Method | CIFAR-10N | | | | | CIFAR-100N |
|---|---|---|---|---|---|---|
| | Aggre | Ran1 | Ran2 | Ran3 | Worst | Noisy |
| Forward | 88.2 | 86.9 | 86.1 | 87.0 | 79.8 | 57.0 |
| JoCoR | 91.4 | 90.3 | 90.2 | 90.1 | 83.3 | 60.0 |
| CORES | 91.2 | 89.7 | 89.9 | 89.8 | 83.6 | 61.2 |
| CO-Teaching | 91.2 | 90.3 | 90.3 | 90.2 | 83.8 | 60.4 |
| CO-Teaching++ | 90.6 | 89.7 | 89.5 | 89.5 | 83.3 | 57.9 |
| FINE | 91.0 | 90.1 | 89.9 | 90.1 | 81.1 | 58.4 |
| BHN | 86.1 | 84.9 | 84.9 | 84.8 | 78.4 | 56.9 |
| RNT | 80.8 | 79.1 | 78.9 | 79.6 | 69.7 | 53.1 |
| **Ours** | **92.3** | **91.6** | **91.3** | **91.2** | **85.6** | **64.7** |

Table 4: Results on CIFAR-10/100 with symmetric and asymmetric label noise.

| Dataset | CIFAR-10 | | | CIFAR-100 | | |
|---|---|---|---|---|---|---|
| **Noisy Type** | Sym | | Asym | Sym | | Asym |
| **Noise Ratio** | 20 | 50 | 40 | 20 | 50 | 40 |
| L2RW (last) | 89.4 | 83.2 | 84.1 | 65.3 | 50.3 | 44.6 |
| L2RW (best) | 90.6 | 85.3 | 86.0 | 67.8 | 55.6 | 50.0 |
| MW-Net (last) | 85.8 | 74.5 | 77.8 | 63.0 | 46.5 | 45.3 |
| MW-Net (best) | 91.2 | 85.6 | 88.6 | 67.5 | 58.2 | 53.4 |
| **Ours (FBR)** | **92.3** | **87.0** | **90.6** | 73.4 | **65.4** | **73.2** |
| **Ours (NTK)** | 91.4 | 86.4 | 89.7 | **73.6** | **65.4** | 73.1 |

# D  IMPLEMENTATION DETAILS AND ADDITIONAL RESULTS

## D.1  IMPLEMENTATION DETAILS

*Architectures and hyperparameters.* Network backbones and all optimization hyperparameters for our method and baselines match those in Kim et al. (2021) to ensure fair comparison. Specifically, we use ResNet-34 models for CIFAR-10 and CIFAR-100. We use the same set of hyper-parameters for CIFAR-10 and CIFAR-100. During training, we set a batch size of 128. We use SGD with a weight decay $5 \times 10^{-4}$ and a momentum of 0.9. The learning rate is initialized as 0.02 and decrease by a factor of 10 at epochs 40, 80 and 100. For Clothing 1M, we compare with SCE, ELR (Liu et al., 2020), DivideMix (Li et al., 2020), CORES (Cheng et al., 2020), FINE (Kim et al., 2021), BHN (Yu et al., 2023), RENT (Bae et al., 2024). We use ResNet-50 with weights pre-trained on ImageNet. The batch size is set to 64. We use SGD with initial learning rate 0.01, momentum 0.9, and weight decay $5 \times 10^{e}-4$. We train the neural network for 10 epochs.

## D.2  ADDITIONAL RESULTS

Comparison of sample selection/reweighting methods on CIFAR-10n/100n are provided in Table 3. For meta-learning, Table 4 reports MW-Net baselines under last-epoch and best-epoch selection. As shown in the table, MW-Net exhibits severe overfitting under label noise.

## D.3  EFFECT OF CLEAN SUBSET SIZE

We additionally conducted experiments with different clean subset sizes. We consider the setting with $40\%$ symmetric noise. The results in Table 6 demonstrate that the test accuracy is not sensitive to the size of the clean subset, remaining stable around $89\%$ when the number of clean samples varies from 1000 to 3000.

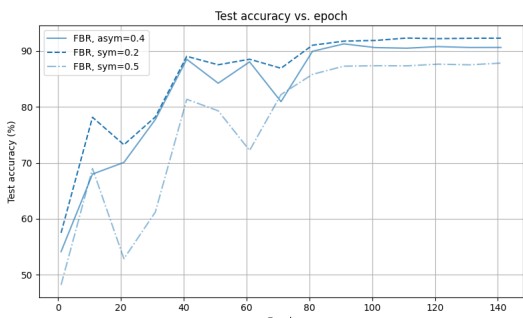

Figure 2: Illustration of robustness to late-stage overfitting to noisy labels at a variety of noise rates

Table 5: Test accuracy (%) on CIFAR-10 with different clean subset sizes.

| Size | 1000 | 1500 | 2000 | 2500 | 3000 |
|---|---|---|---|---|---|
| Accuracy | 89.4 | 89.6 | 89.6 | 89.8 | 89.6 |

## D.4 HIGH NOISE RATE

We further evaluate our method under an extremely noisy setting with 90% symmetric label noise. As shown in Table 6, our approach achieves a test accuracy of 58.5% on CIFAR-10 in this challenging scenario.

Table 6: Test accuracy (%) on CIFAR-10 and CIFAR-100 under 90% symmetric label noise.

|  | CIFAR-10 | CIFAR-100 |
|---|---|---|
| Accuracy | 58.5 | 19.9 |

## D.5 INSTANCE NOISE

Following the instance-dependent noise generation protocol of Cheng et al. (2020), we synthesize label noise on CIFAR-10 and CIFAR-100, where each sample is flipped according to an instance-specific transition distribution. We then evaluate our method under this setting. As reported in Table 7, our approach attains test accuracies of 91.0% on CIFAR-10 and 69.5% on CIFAR-100 at a 20% noise rate, and it remains strong even when the noise ratio is increased to 40%.

Table 7: Test accuracy (%) on CIFAR-10 and CIFAR-100 under instance-dependent label noise.

| Dataset | CIFAR-10 | | CIFAR-100 | |
|---|---|---|---|---|
| Noise Ratio | 20 | 40 | 20 | 40 |
| Accuracy | 91.0 | 81.5 | 69.5 | 62.0 |

## D.6 NOISY SUBSET

We also conducted an experiment under the same noise setting on the *clean subset* with its size fixed to 2000 under 40% symmetric noise. Our algorithm achieves a test accuracy of 88.7% in this setting, which is slightly lower than the 89.6% obtained when using the clean subset.

## D.7 VARIOUS NEURAL NETWORK ARCHITECTURES

In this subsection, we demonstrate that our noisy-label elimination strategy is effective across different neural network architectures. Specifically, we conduct experiments with a PreAct-ResNet-18

backbone, a deeper ResNet-101, and CIFAR-style ResNet-56 and CIFAR-ResNet-110, thus covering both shallower and deeper residual networks commonly used on CIFAR benchmarks. We visualize the terminal weight histogram and the empirical distribution of negative weight derivatives in Fig. 3, and report the corresponding test accuracies in Table 8.

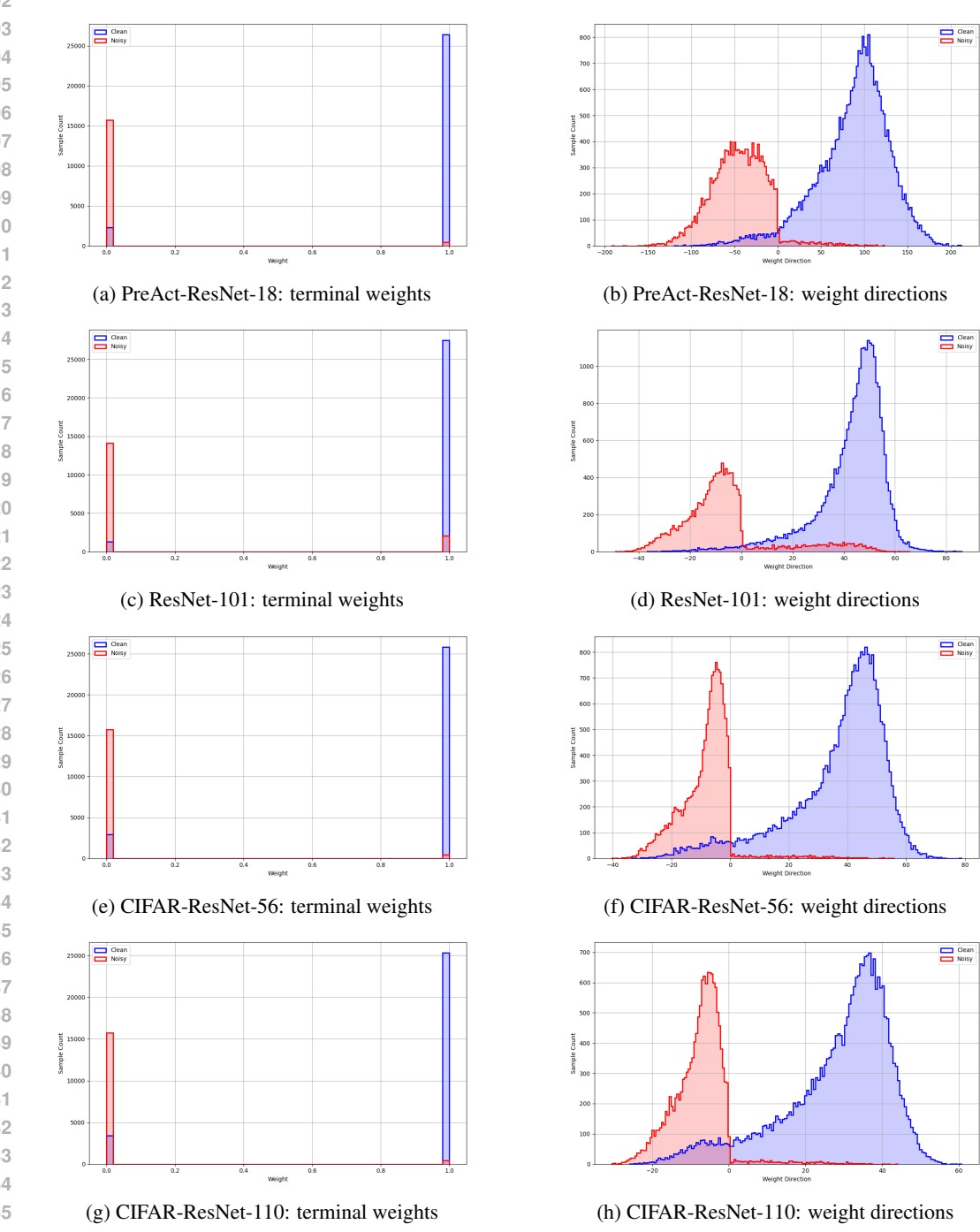

(a) PreAct-ResNet-18: terminal weights

(b) PreAct-ResNet-18: weight directions

(c) ResNet-101: terminal weights

(d) ResNet-101: weight directions

(e) CIFAR-ResNet-56: terminal weights

(f) CIFAR-ResNet-56: weight directions

(g) CIFAR-ResNet-110: terminal weights

(h) CIFAR-ResNet-110: weight directions

Figure 3: Terminal weight histograms and empirical distributions of negative weight derivatives for different architectures. Each row corresponds to one architecture (from top to bottom: PreAct-ResNet-18, ResNet-101, CIFAR-ResNet-56, CIFAR-ResNet-110); left: terminal weight histogram, right: empirical distribution of negative weight derivatives.

Table 8: Test accuracy (%) of different architectures on CIFAR-10 under $40\%$ symmetric label noise.

| Architecture | Accuracy |
|---|---|
| PreAct-ResNet-18 | 87.2 |
| ResNet-101 | 85.2 |
| CIFAR-ResNet-56 | 86.0 |
| CIFAR-ResNet-110 | 84.3 |

## D.8 CLASS IMBALANCE

Following the standard long-tailed CIFAR protocol (Zhang et al., 2022), we construct a long-tailed version of CIFAR-10 with imbalance ratio $\rho = 50$ by reducing the number of training samples in each class according to an exponential decay: $n_k = n_{\max}\rho^{-k/(C-1)}$ for class $k$. On top of this long-tailed training set, we inject class-independent symmetric label noise with rate $\eta = 0.2$. The terminal weight histogram and the empirical distribution of negative weight derivatives are visualized in Figure 4.

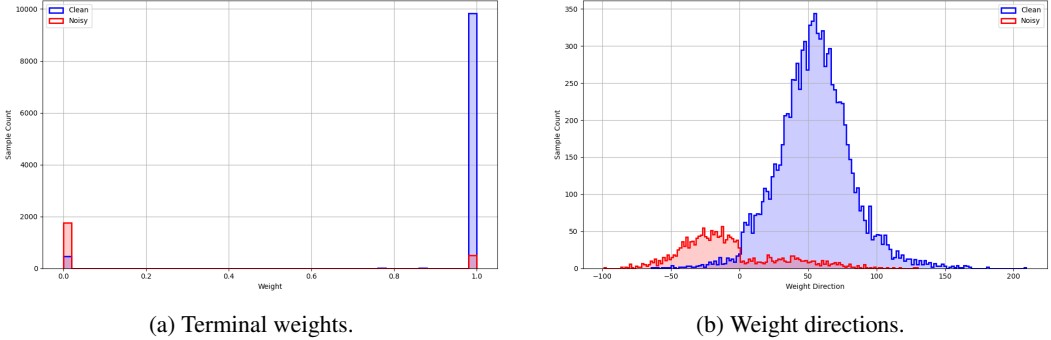

(a) Terminal weights.         (b) Weight directions.

Figure 4: Class Imbalance.

## D.9 OPEN SET NOISE

In the open-set noise setting, we corrupt the CIFAR-10 training set by replacing $20\%$ of the training images with out-of-distribution images sampled from CIFAR-100 and assigning each such image a random label from $\{0, \dots, 9\}$. Our algorithm achieves $92.7\%$ test accuracy under this challenging setting, demonstrating strong robustness to open-set noisy labels.

## D.10   NTK SEPARATION ASSUMPTIONS ON CIFAR

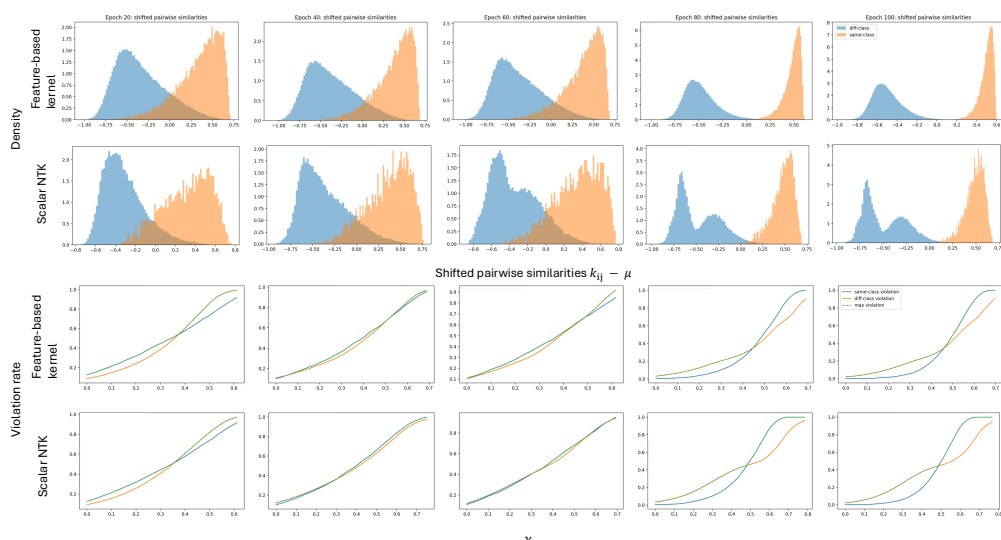

Figure 5: Evaluating assumption 3 and 4 on CIFAR-10 using both an NTK-inspired kernel and a feature-based kernel. **Top:** Separation of kernel values for randomly sampled pairs of data points that share the class and differ in class. **Bottom:** For the same set of randomly sampled points, line plots of the violation rate of the assumption over different values of $\gamma$.

In Figure 5, we demonstrate two experiments using CIFAR-10 to probe our paper's kernel assumptions (assumptions 3 and 4). In the first, "feature-based" experiment, we take a ResNet-34 and extract mean-centered penultimate-layer features for a subset of training images, and define a scalar kernel $K(x_i, x_j)$ as the normalized, shifted inner product of those features. This aligns with the method we introduce in the main text. We then sample same-class and different-class pairs and plot the histograms of their kernel values, and compute margin-violation curves $\gamma \mapsto$ (fraction of same-class pairs with $K < \gamma$, fraction of different-class pairs with $K > -\gamma$), which tests the separation/margin assumption on CIFAR-10. In the second, NTK-inspired experiment, we replace features by per-logit gradient features $\phi_c(x) = \nabla_{W_{\text{pen}}} f_c(x)$ w.r.t. the penultimate weights and define non-symmetric and symmetric scalar kernels on training indices (e.g. $K(i, j) = \langle \phi_{y_i}(x_i), \phi_{y_i}(x_j) \rangle$). We can see how the assumptions hold both in the feature regime and in an NTK-like gradient regime as the network trains on CIFAR-10.

## D.11   WEIGHT DYNAMICS OF FBR

We plot the weight dynamics under the same setting as in Figure 1a. As shown in Figure 6, the weight dynamics of FBR are noticeably more stable.

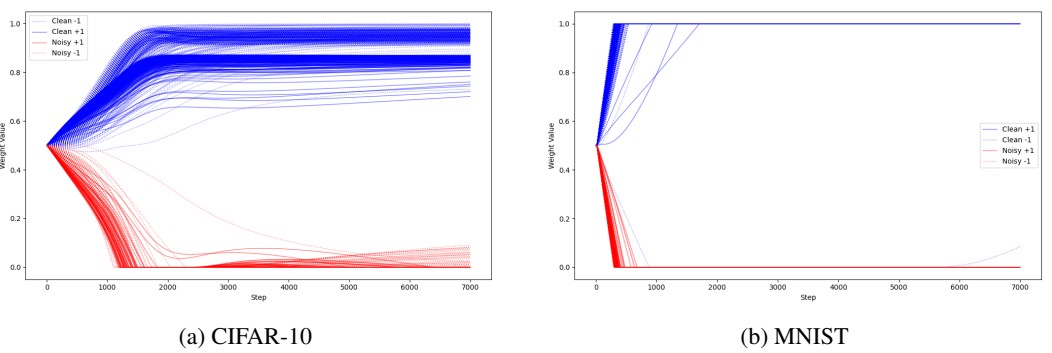

(a) CIFAR-10                                      (b) MNIST

Figure 6: Weight dynamics comparison.

## D.12 Additional experiments on Mini-WebVision

Here, we provide a new experiment. All methods in Table 9 are trained on the standard mini-WebVision benchmark (50-class subset of WebVision) using only the noisy training split, and evaluated on the official clean mini-WebVision validation set in terms of Top-1 and Top-5 accuracy. Following prior work (CRUST, FINE, ELR/ELR+), all results use an Inception-ResNet-v2 backbone.

As shown in Table 9, earlier noisy-label baselines such as F-correction, Decoupling, Co-teaching, MentorNet, D2L, and INCV achieve Top-1 accuracies in the low- to mid-60% range. More recent sample-selection and robust-training methods improve performance substantially: CRUST reaches 72.40%/89.56% Top-1/Top-5, FINE reaches 75.24%/90.28%, and ELR+ further improves to 77.78%/91.68%. Our method attains 80.01% Top-1 and 92.13% Top-5, outperforming FINE by about +4.8 Top-1 points and ELR+ by about +2.2 Top-1 points, and improving over CRUST by roughly +7.6 Top-1 and +2.6 Top-5.

| Mini-WebVision | Top-1 (%) | Top-5 (%) |
|---|---|---|
| F-correction | 61.12 | 82.68 |
| Decoupling | 62.54 | 84.74 |
| Co-teaching | 63.58 | 85.20 |
| MentorNet | 63.00 | 81.40 |
| D2L | 62.68 | 84.00 |
| INCV | 65.24 | 85.34 |
| CRUST | 72.40 | 89.56 |
| FINE | 75.24 | 90.28 |
| ELR | 76.26 | 91.26 |
| ELR+ | 77.78 | 91.68 |
| Ours | **80.01** | **92.13** |

Table 9: Top-1 and Top-5 accuracy (%) on mini-WebVision.

## E Analysis of Computational Cost

**A note on computational cost of FBR, L2R, and MWN.** FBR's reweighting step operates entirely in feature space and adds a single Gram matrix multiplication plus cheap row-wise operations whose cost is $\mathcal{O}(Bmd)$ per mini-batch and independent of classifier depth. Below, we discuss in more detail how FBR is substantially cheaper than existing meta-reweighting methods, especially for large models where additional full forward/backward passes dominate the runtime.

**Computational cost of feature-base reweighting** Let $B$ denote the batch size (number of training samples reweighted per step), $m$ the number of clean/validation samples, and $d$ the feature dimension (e.g., penultimate-layer representation size). The dominant operation in FBR is the batch–clean Gram matrix computation:

$$K_B = \tilde{\Phi}_B \tilde{\Phi}_v^\top \in \mathbb{R}^{B \times m},$$

where $\tilde{\Phi}_B \in \mathbb{R}^{B \times d}$ and $\tilde{\Phi}_v \in \mathbb{R}^{m \times d}$ are mean-centered feature matrices. The per-batch time and memory complexity of the reweighting step are therefore:

$$T_{\text{reweight}}(B, m, d) = O(Bmd), \quad M_{\text{reweight}}(B, m, d) = O(Bd + md + Bm),$$

corresponding to (i) computing the Gram matrix, and (ii) storing features and the $B \times m$ similarity matrix.

**Computational cost of L2R and MWN** Learning-to-reweight (L2R) incurs a roughly $3\times$ slowdown over standard SGD: as the authors state, the procedure "requires two full forward and backward passes of the network on training and validation respectively, and then another backward on backward pass . . . and finally a backward pass to minimize the reweighted objective," leading them

to conclude that "compared to regular training, our method needs approximately $3\times$ training time." Meta-Weight-Net (MWN) exhibits a comparable constant-factor overhead: although it replaces implicit second-order steps with an explicit MLP $V(\cdot; \Theta)$ that maps per-example losses to weights, each iteration still requires a forward pass through V, at least one classifier forward–backward pass on the training batch to form $\hat{w}^{(t)}(\Theta)$, another on the meta batch to compute the meta-gradient, and a final classifier update using the current weights.

## F  THE USE OF LARGE LANGUAGE MODELS

We used a large language model (ChatGPT) only to aid with grammar, wording, and stylistic polishing of text. All ideas, results, and claims are our own; we manually verified factual statements and citations. Only non-sensitive draft text was provided to the tool, and all outputs were reviewed and edited by the authors. Any remaining errors are our responsibility.

