# OpenReview forum: "Revisiting Meta-Learning with Noisy Labels: Reweighting Dynamics and Theoretical Guarantees"
_ICLR.cc/2026/Conference — Submitted to ICLR 2026_

### Official Review · Reviewer_WxuT · 2025-10-26

**Soundness:** 3
**Presentation:** 3
**Contribution:** 3
**Rating:** 6
**Confidence:** 4

**Summary:**

This paper studies meta-learning based sample reweighting approaches for learning with noisy labels. The authors note that overparameterized networks easily memorize corrupted labels, and while meta-reweighting methods (which use a small clean validation set to guide weights for training samples) are empirically effective, their behaviour and theoretical dynamics remain under-explored. They provide a rigorous theoretical analysis of the training trajectory of meta-reweighting under label noise, showing it decomposes into three phases: (i) alignment – clean examples are amplified and conflicting noisy examples suppressed; (ii) filtering – weights of noisy examples go to near zero until the clean-subset loss plateaus; and (iii) post-filtering – when the clean subset loss is very small, the coupling term vanishes and discrimination becomes perturbation-sensitive. Based on these insights, they propose a lightweight surrogate algorithm for meta-reweighting (incorporating mean-centering, row-shifting, label-signed modulation) that avoids expensive bilevel optimization yet achieves improved robustness. They validate on synthetic and real noisy-label benchmarks and show superior performance over baselines.

**Strengths:**

Deep theoretical insight: The authors bridge the gap between empirical success of meta-reweighting and rigorous understanding of its dynamics.

Practical algorithmic outcome: Deriving a surrogate algorithm from theory is valuable for practitioners dealing with noisy data.

Clear experiments: The paper presents empirical results validating the theoretical claims and demonstrating improvement over baselines.

**Weaknesses:**

Empirical scope: More diverse architectures (e.g., large scale deep networks), more varied noise types (instance‐dependent noise, real-world label errors), and more explicit ablation of assumption breakdown would enhance confidence.

Practical cost/complexity: While the surrogate avoids bilevel optimization, the paper could provide more detail on computational cost, scalability, and comparison to simpler baselines (e.g., early stopping, loss clipping).

Clarification of limitations: More explicit discussion of when and why the method might fail or degrade (e.g., extremely high noise rates, unbalanced classes, small clean subset) would benefit readers.

**Questions:**

How sensitive is the surrogate algorithm to the size of the clean subset (meta-validation set)? If the clean subset is very small or itself noisy, how does performance degrade?

Could you provide more detailed runtime/compute cost comparisons between the surrogate algorithm and full bilevel meta-reweighting, across different network sizes and noise regimes?

How does the method perform under more challenging noise regimes (e.g., instance‐dependent noise, highly imbalanced labels, open‐set noise) beyond the synthetic or standard benchmarks?

---

> ### Author Response · Authors · 2025-11-24
> **Author Response (Part 1/2)**
>
> We thank the reviewer for recognizing our theoretical and algorithmic contribution and for their thoughtful and detailed comments about the empirical scope of our paper. We also appreciate the reviewer for recognizing our theoretical and algorithmic contribution. Below, we address questions regarding the experimental scope, ablation experiments, and computational cost and look forward to further discussion. We would be happy to provide additional clarification or run additional experiments during the rebuttal phase.
>
> **New experiment on mini-webvision.**
> In the response to all reviewers, we include a new experiment using a large-scale dataset with realistic noise. We evaluate our method on the mini-WebVision dataset using an Inception-ResNet v2 network. Among score-based methods, we demonstrate state-of-the-art results by a significant margin with respect to both top-1 and top-5 accuracy. We encourage the reviewer to check them out.
>
> **Sensitivity of kernel-based reweighting to the size and quality of the meta subset.**
> In the revised version of the manuscript, we include two ablation studies measuring the affect of meta subset noise and size on the quality of the final predictor (Appendix D.3 and D.6).
>
> To highlight these results, on CIFAR-10 with the meta-subset size fixed to $2000$ (the same as the main-text experiments) and with meta-subset labels perturbed by $40\%$ symmetric noise, our algorithm achieves a test accuracy of $88.7\%$, which is slightly lower than the $89.6\%$ obtained when using the clean subset.
> Below, we also explore how the test accuracy varies as the size of the clean meta subset varies, while keeping the training label noise fixed with 40\% symmetric noise. It is interesting to note the surprising resilience of our method to varying this meta subset.
>
> **Table: Effect of meta set size on accuracy.**
>
> | Meta set size | 1000 | 1500 | 2000 | 2500 | 3000 |
> |--------------:|-----:|-----:|-----:|-----:|-----:|
> | Accuracy      | 89.4 | 89.6 | 89.6 | 89.8 | 89.6 |
>
>
> **Instance Noise** (Appendix D.5)
> We synthesize label noise on CIFAR-10 and CIFAR-100, where each sample is flipped according to an instance-specific transition distribution. We then evaluate our method under this setting. As reported in Table 7 in our revised submission, our approach attains test accuracies of $91.0\%$ on CIFAR-10 and $69.5\%$ on CIFAR-100 at a $20\%$ noise rate, and it remains strong even when the noise ratio is increased to $40\%$.
>
> **Table: Instance-dependent noise experiments on CIFAR-10 and CIFAR-100.**
>
> | Dataset      | CIFAR-10 | CIFAR-10 | CIFAR-100 | CIFAR-100 |
> |--------------|---------:|---------:|----------:|----------:|
> | Noise ratio  | 20       | 40       | 20        | 40        |
> | Accuracy     | 91.0     | 81.5     | 69.5      | 62.0      |
>
>
> **Open-set noise** (Appendix D.9)
> In the open-set noise setting, we corrupt the CIFAR-10 training set by replacing
> $20\%$ of the training images with out-of-distribution images sampled from
> CIFAR-100 and assigning each such image a random label from $\{0,\dots,9\}$.
> Our algorithm achieves $92.7\%$ test accuracy under this challenging setting,
> demonstrating strong robustness to open-set noisy labels.
>
> **Performance with alternative architectures.** In Appendix D.7 of the revision, we provide a few new experiments on additional architectures. To verify that our noisy-label elimination strategy is not tied to a specific
> backbone, we also evaluate our method with PreAct-ResNet-18, ResNet-101, CIFAR-ResNet-56,
> and CIFAR-ResNet-110, covering both shallow and deep residual networks on
> CIFAR-10. As shown in Figure 3 of the revised manuscript, our method consistently separates noisy labels from clean ones across all
> these architectures, indicating that the underlying reweighting mechanism
> transfers robustly to different network designs.
>
> **Additional experiments.** Additional experiments with extremely high noise rates and unbalanced classes are also discussed in the response to all reviewers. We also included the results in Appendix sections D.4 and D.8 of our revised submission.

---

> > ### Author Response · Authors · 2025-11-24
> > **Author Response (Part 2/2)**
> >
> > **A note on computational cost of FBR, L2R, and MWN.**
> >
> > FBR’s reweighting step operates entirely in feature space and adds a single Gram matrix multiplication plus cheap row-wise operations whose cost is $\mathcal{O}(Bmd)$ per mini-batch and independent of classifier depth. Below, we discuss in more detail how FBR is substantially cheaper than existing meta-reweighting methods, especially for large models where additional full forward/backward passes dominate the runtime.
> >
> > *Computational cost of feature-base reweighting.* Let $B$ denote the batch size (number of training samples reweighted per step), $m$ the number of clean/validation samples, and $d$ the feature dimension (e.g., penultimate-layer representation size). The dominant operation in FBR is the batch–clean Gram matrix computation:
> >
> > $$
> > K_B = \tilde\Phi_B \tilde\Phi_v^\top \in \mathbb{R}^{B \times m},
> > $$
> > where $\tilde\Phi_B \in \mathbb{R}^{B \times d}$ and $\tilde\Phi_v \in \mathbb{R}^{m \times d}$ are mean-centered feature matrices.
> > The per-batch time and memory complexity of the reweighting step are therefore:
> > $$
> > T_{\text{reweight}}(B,m,d) = O(B m d),\quad
> > M_{\text{reweight}}(B,m,d) = O(B d + m d + B m),
> > $$
> > corresponding to (i) computing the Gram matrix, and (ii) storing features and the $B \times m$ similarity matrix.
> >
> > *Computational cost of L2R and MWN.*
> > Learning-to-reweight (L2R) incurs a roughly $3\times$ slowdown over standard SGD: as the authors state, the procedure “requires two full forward and backward passes of the network on training and validation respectively, and then another backward on backward pass … and finally a backward pass to minimize the reweighted objective,” leading them to conclude that “compared to regular training, our method needs approximately $3\times$ training time.” Meta-Weight-Net (MWN) exhibits a comparable constant-factor overhead: although it replaces implicit second-order steps with an explicit MLP $V(\cdot;\Theta)$ that maps per-example losses to weights, each iteration still requires a forward pass through V, at least one classifier forward–backward pass on the training batch to form $\hat w^{(t)}(\Theta)$, another on the meta batch to compute the meta-gradient, and a final classifier update using the current weights.

---

### Official Review · Reviewer_irZ1 · 2025-10-29

**Soundness:** 3
**Presentation:** 3
**Contribution:** 2
**Rating:** 4
**Confidence:** 3

**Summary:**

In this paper, the authors conduct a detailed theoretical analysis to reveal the mechanism of the meta-reweighting-based methods for training with label noise. As shown in the results, the training process can be mainly divided into three stages: an alignment stage, a filtering stage, and a post-filtering stage. Based on the theoretical study, the authors propose a lightweight surrogate for meta reweighting and demonstrate its effectiveness on synthetic and real noise datasets.

**Strengths:**

1. The theoretical analysis in Section 3 provides a reasonable explanation for the training mechanism of the meta-reweighting framework for the first time.

2. The proposed method, FBR, is a lightweight surrogate for the meta reweighting update, reducing the computational complexity.

**Weaknesses:**

1. Though the theoretical analysis is good under the settings the authors focus on, it is not comprehensive enough regarding the following aspects:

- The analysis only focuses on the squared loss, but the cross-entropy loss is more widely used for the classification task, which has not been discussed in this work.

- The main tool used for the analysis is the neural tangent kernel (NTK). However, NTK approximation requires strong assumptions about the deep models, which might not hold in practice.

- It seems that the analysis is mainly for the L2RW method without an explicit weighting function, and it is unclear whether the results still hold for MW-Net.

2. The experiments have some inefficiencies:

- There are no studies on the convergence behaviors of the proposed method.

- As mentioned before, the NTK approximation might deviate from the neural networks in practice, and thus it would be useful to test networks with different hyperparameters.

**Questions:**

The authors should address the issues mentioned in the "Weaknesses".

---

> ### Author Response · Authors · 2025-11-24
> **Author Response (Part 1/2)**
>
> We thank the reviewer for recognizing our contribution and for their thoughtful and detailed comments. We appreciate the reviewer for recognizing our theoretical and algorithmic contribution. Below, we address questions and look forward to further discussion. We would be happy to provide additional clarification or run additional experiments during the rebuttal phase.
>
> **Bridging squared-loss and multi-class cross entropy.**
>
> We provide a more comprehensive discussion in the response to all reviewers. Briefly, our proposed method can be viewed as decoupling sample reweighting from the
> subsequent training step. The reweighting signals are obtained
> using a binary MSE-style surrogate (via a one-vs-all construction) that
> separates clean and noisy samples. Then the resulting per-example
> weights can be used in an arbitrary supervised training loss. From a theoretical perspective, using the MSE is standard in the literature on deep learning theory, as MSE-type objectives admit tractable kernel-based characterizations (see, e.g., [1,2]). There is also work showing that, in certain overparameterized regimes, the choice of training loss can have little effect on the resulting interpolating solution [3]. Extending our framework to directly analyze meta-reweighting under a multiclass cross-entropy objective would require a substantially different technical setup, which is beyond the scope of this work. Whether such an analysis would ultimately produce a reweighting signal that is provably superior to the MSE-based one we derive here is an interesting open question, and we view it as an important direction for future work. We will clarify this discussion in the revised manuscript.
>
> **NTK assumption may not hold in practice.**
>
> The NTK regime together with Assumption 3 (or other kernel separability assumptions) provides a *sufficient* condition under which a good reweighting signal is maintained over training. In more complicated settings, the kernel values may not remain constant if the neural network does not satisfy the NTK assumptions or if the training dynamics are driven by cross-entropy loss rather than the squared loss considered in our theory. Our experiments in Appendix D.10 show that this separability property persists empirically even when the kernel dynamics deviate from the idealized NTK regime or when the model is trained with multiclass cross-entropy.
> We run two CIFAR-10 experiments to probe our paper’s kernel assumptions (assumptions 3 and 4). In the first, ``feature-based" experiment, we take a ResNet-34 and extract mean-centered penultimate-layer features for a subset of training images, and define a scalar kernel $K(x_i,x_j)$ as the  normalized, shifted inner product of those features. This aligns with the practical method we introduce in the paper. We then sample same-class and different-class pairs and plot the histograms of their kernel values, and compute margin-violation curves $\gamma \mapsto \text{(fraction of same-class pairs with } K<\gamma, \text{ fraction of different-class pairs with } K>-\gamma)$, which tests the separation/margin assumption on CIFAR-10. In the second, NTK-inspired experiment, we replace features by per-logit gradient features $\phi_c(x) = \nabla_{W_{\text{pen}}} f_c(x)$ w.r.t. the penultimate weights and define non-symmetric and symmetric scalar kernels on training indices (e.g. $K(i,j)=\langle\phi_{y_i}(x_i),\phi_{y_i}(x_j)\rangle$. We can see how the assumptions hold both in the feature regime and in an NTK-like gradient regime as the network trains on CIFAR-10.
>
> **Analysis of L2RW and MW-Net.**
>
> Our analysis is carried out for a generic meta-reweighting update where the
> per-example weights $w$ are optimized by gradient descent on the validation
> loss (Eq. (1)).
> MW-Net can be viewed as a smooth reparametrization of the same scheme, where
> the weights are given by $w_i = V(\ell_i(\theta);\Theta)$ and the meta-gradient
> with respect to $\Theta$ is obtained via the chain rule
> $\nabla_{\Theta} L_{\text{val}} = (\partial w/\partial \Theta)^{\top}
> \nabla_w L_{\text{val}}$.
> Thus, the sample-wise signal that drives the evolution of weights is still
> $\nabla_w L_{\text{val}}$. A fully rigorous extension to arbitrary parametric weighting networks is
> technically involved, and we leave this as an interesting direction for
> future work.
>
> [1] Hu T, Wang J, Wang W, Li Z. Understanding square loss in training overparametrized neural network classifiers. Advances in Neural Information Processing Systems. 2022 Dec 6;35:16495-508.
>
> [2] Seleznova M, Weitzner D, Giryes R, Kutyniok G, Chou HH. Neural (tangent kernel) collapse. Advances in Neural Information Processing Systems. 2023 Dec 15;36:16240-70.
>
> [3] Muthukumar V, Narang A, Subramanian V, Belkin M, Hsu D, Sahai A. Classification vs regression in overparameterized regimes: Does the loss function matter?. Journal of Machine Learning Research. 2021;22(222):1-69.

---

> > ### Author Response · Authors · 2025-11-24
> > **Author Response (Part 2/2)**
> >
> > **Convergence behavior of kernel-based reweighting.**
> >
> > In the table below, we demonstrate significant improvements in the converged solution compared to an alternative reweighting strategy (Meta-Weight-Net). For 3 different noise models on CIFAR-10, we identify the "best" test-error, which typically occurs early during the training process and the "final" test-error, which occurs after 150 epochs. In this table, we emphasize the robustness of FBR to overfitting noisy labels (depicted by the small $\Delta$ between the best and final accuracies). The convergence plot can also be found in Figure 2 of our revised submission.
> >
> > **Table:** Test accuracy on CIFAR-10 under different label noise settings. “Best” denotes the maximum test accuracy over training, and “Final” the accuracy at the last epoch, with the change (Final − Best) in parentheses. For each noise setting, the better method is bolded.
> >
> > | Noise type | Noise rate | FBR Best | FBR Final (Δ)     | MW Best | MW Final (Δ)      |
> > |-----------|-----------:|---------:|-------------------:|--------:|-------------------:|
> > | asym      | 0.4        | **91.23** | **90.50 (-0.73)** | 89.88   | 77.79 (-12.09)    |
> > | sym       | 0.2        | **92.35** | **92.15 (-0.20)** | 91.26   | 85.81 (-5.45)     |
> > | sym       | 0.5        | **87.80** | **87.51 (-0.29)** | 86.56   | 74.51 (-12.05)    |
> >
> >
> > **Performance with alternative network hyperparameters.**
> >
> > To verify that our noisy-label elimination strategy is not tied to a specific
> > backbone, in Appendix D.7 we also evaluate it with PreAct-ResNet-18, ResNet-101, CIFAR-ResNet-56,
> > and CIFAR-ResNet-110, covering both shallow and deep residual networks on
> > CIFAR-10. As shown in Figure 3 of the revised manuscript, our method consistently separates noisy labels from clean ones across all
> > these architectures, indicating that the underlying reweighting mechanism
> > transfers robustly to different network designs.

---

### Official Review · Reviewer_1Tx9 · 2025-10-30

**Soundness:** 1
**Presentation:** 2
**Contribution:** 1
**Rating:** 0
**Confidence:** 5

**Summary:**

The paper studies training with noisy labels and finds that validation-based sample reweighting overfits late in training: noisy samples get higher weights. It analyzes why this happens and proposes a lightweight surrogate that avoids bilevel optimization to mitigate the issue. The method is evaluated on several noisy-label benchmark datasets.

**Strengths:**

- The writing and structure are relatively clear.
- The focus on the increasing weights of noisy samples in the later stages of training is interesting and practically relevant.

**Weaknesses:**

- Unrealistic and unjustified theoretical setup: The paper conducts its theoretical analysis in a binary classification setting and adopts the mean squared error (MSE) as the loss function. This choice is questionable: MSE is not suitable for classification tasks. More importantly, the experiments are performed under multi class classification with cross entropy loss, which makes the theoretical results difficult to generalize or connect to the empirical part. The authors provide no justification for this setup and make no attempt to relate the theory to the experimental conditions.
- Insufficient rigor and validation in explaining the key mechanism: The authors attribute late stage overfitting to “approximation error.” While this explanation is intuitively plausible, the paper lacks any systematic analysis of when and under what conditions this phenomenon occurs. For instance, overfitting of this kind is often not observed under large learning rates—can the proposed theory explain this behavior? The current discussion is coarse and lacks a quantitative relationship between key variables (e.g., learning rate, training stage) and various sources of error(e.g. approximation error, estimation error). No empirical verification is provided, such as curves of approximation error over time or sensitivity to learning rate. As a result, the core mechanism remains speculative and weakly supported.
- Main claims are not supported by direct evidence: The paper claims that the proposed method effectively avoids late stage overfitting, yet it only shows weight dynamics of other methods (L2W) and omits those of the proposed approach. If the ability to avoid weight overfitting is the main contribution, this missing evidence is critical. Furthermore, the method is described as a lightweight surrogate of L2W, but no complete L2W baseline is included in the experiments. It is thus unclear whether the reported improvement truly results from avoiding overfitting or from other unreported strategies or hyperparameter adjustments. If additional techniques are involved, ablation studies are essential to verify which component actually contributes to the improvement.
- Empirical evaluation is far below current community standards：The experimental evaluation is highly limited. The paper only tests on a few datasets and includes very few baselines. It does not consider instance dependent noise, nor does it evaluate on large scale real noisy datasets such as WebVision. The comparison set includes only a few classic baselines and omits recent SOTA methods such as Modified-DivideMix[1], L2B[2] and DCD[3]. Consequently, the experimental evidence is not convincing and would not meet the current standards in the noisy label learning or meta reweighting literature.
The phenomenon addressed in this paper is practically relevant, but the theoretical setup is oversimplified and disconnected from the experiments, the main explanation lacks rigorous validation, and the empirical evaluation falls well below the expectations of the field. Overall, the work does not provide sufficient evidence to support its claims.

[1] Yuan S, Li X, Miao Y, et al. Combating noisy labels by alleviating the memorization of dnns to noisy labels[J]. IEEE Transactions on Multimedia, 2024.

[2] Zhou Y, Li X, Liu F, et al. L2B: Learning to bootstrap robust models for combating label noise[C]. Proceedings of the IEEE/CVF Conference on Computer Vision and Pattern Recognition. 2024: 23523-23533.

[3] Mu C, Qu Y, Yan J, et al. Meta-Learning Dynamic Center Distance: Hard Sample Mining for Learning with Noisy Labels[C]. Proceedings of the IEEE/CVF International Conference on Computer Vision. 2025: 415-425.

**Questions:**

See the weakness

---

> ### Author Response · Authors · 2025-11-24
> **Author Response (Part 1/3)**
>
> We thank the reviewer for their detailed comments. We appreciate the reviewer recognizing our efforts in composing the manuscript. Below, we address questions and look forward to further discussion. We would be happy to provide additional clarification or run additional experiments during the rebuttal phase and look forward to discussing our submission with you further during the discussion phase. Please let us know!
>
> **Clarification of contribution.**
>
> We would like to briefly summarize the contribution of our work and also respond to a few of the weaknesses outlined by the reviewer. In our paper, we provide a rigorous analysis of meta-reweighting dynamics, leading to a three-phase characterization that clarifies when and why meta-reweighting succeeds or fails. *Experimentally, we consider only score-based methods or methods which do not re-assign or correct mislabeled examples*. We see our method as complementary to techniques that either post-process scores to produce relabeled examples or utilize SSL-based techniques.
>
> **Bridge between analysis in the binary multiclass regime and multi-class cross-entropy.**
>
> We provide a more comprehensive discussion in the response to all reviewers. Briefly, our proposed method can be viewed as decoupling sample reweighting from the
> subsequent training step. The reweighting signals are obtained
> using a binary MSE-style surrogate (via a one-vs-all construction) that
> separates clean and noisy samples. Then the resulting per-example
> weights can be used in an arbitrary supervised training loss. From a theoretical perspective, using the MSE is standard in the literature on deep learning theory, as MSE-type objectives admit tractable kernel-based characterizations (see, e.g., [1,2]). There is also work showing that, in certain overparameterized regimes, the choice of training loss can have little effect on the resulting interpolating solution [3]. Extending our framework to directly analyze meta-reweighting under a multiclass cross-entropy objective would require a substantially different technical setup, which is beyond the scope of this work. Whether such an analysis would ultimately produce a reweighting signal that is provably superior to the MSE-based one we derive here is an interesting open question, and we view it as an important direction for future work. We will clarify this discussion in the revised manuscript.
>
> [1] Hu T, Wang J, Wang W, Li Z. Understanding square loss in training overparametrized neural network classifiers. Advances in Neural Information Processing Systems. 2022 Dec 6;35:16495-508.
>
> [2] Seleznova M, Weitzner D, Giryes R, Kutyniok G, Chou HH. Neural (tangent kernel) collapse. Advances in Neural Information Processing Systems. 2023 Dec 15;36:16240-70.
>
> [3] Muthukumar V, Narang A, Subramanian V, Belkin M, Hsu D, Sahai A. Classification vs regression in overparameterized regimes: Does the loss function matter?. Journal of Machine Learning Research. 2021;22(222):1-69.

---

> > ### Author Response · Authors · 2025-11-24
> > **Author Response (Part 2/3)**
> >
> > **Late stage overfitting \& approximation error.**
> >
> > We would like to clarify that our theoretical results only provide
> > precise control of the meta-reweighting dynamics in the first two
> > phases (alignment and filtering), where the clean subset loss is still
> > non-negligible and the reweighting signal is well aligned with the
> > clean labels.
> > In the third phase, when the  clean subset loss has essentially
> > converged, we explicitly lose
> > control over the reweighting signals. Our theory therefore does not
> > formally claim that overfitting must occur, but rather explains why
> > such late-stage degradation can arise, as also illustrated by the
> > performance drop in the last training stage in Table 4. A quantitative treatment of how learning rates, training stage, and
> > different sources of error (e.g., approximation vs. estimation)
> > jointly affect generalization is an important but challenging
> > direction, and is orthogonal to our main goal, which is to understand
> > the weight dynamics of meta-reweighting under label noise. We will clarify these limitations and the scope of our claims in the revised version.
> >
> > **Overfitting and weight dynamics of kernel-based reweighting.**
> >
> > We provide results which illustrate the weight dynamics in the late stage of kernel-based reweighting over various architectures (Figure 3 in revision), more complex noise models (Figure 4 in revision). Next, we empirically illustrate robustness to overfitting. In the tables below, we demonstrate significant improvements in the converged solution compared to an alternative reweighting strategy (Meta-Weight-Net). For 3 different noise models on CIFAR-10, we identify the "best" test-error, which typically occurs early during the training process and the "final" test-error, which occurs after 150 epochs. In this table, we emphasize the robustness of FBR to overfitting noisy labels (depicted by the small $\Delta$ between the best and final accuracies). We clarify that this overfitting to noisy labels of meta reweighting is *not* claimed as a theoretical guarantee by our analysis. Rather, we evaluate this phenomenon empirically, as suggested by the reviewer. MW-Net paper [4] already demonstrates consistent improvements over L2RW under
> > comparable settings, and both methods instantiate the same underlying
> > bi-level reweighting scheme. Reproducing a full L2RW baseline in our setup would therefore largely duplicate existing comparisons, so we instead focus on MW-Net as a stronger and widely used representative of L2RW-style meta-reweighting.
> >
> > **Table:** Test accuracy on CIFAR-10 under different label noise settings. “Best” denotes the maximum test accuracy over training, and “Final” the accuracy at the last epoch, with the change (Final − Best) in parentheses. For each noise setting, the better method is bolded.
> >
> > | Noise type | Noise rate | FBR Best | FBR Final (Δ)     | MW Best | MW Final (Δ)      |
> > |-----------|-----------:|---------:|-------------------:|--------:|-------------------:|
> > | asym      | 0.4        | **91.23** | **90.50 (-0.73)** | 89.88   | 77.79 (-12.09)    |
> > | sym       | 0.2        | **92.35** | **92.15 (-0.20)** | 91.26   | 85.81 (-5.45)     |
> > | sym       | 0.5        | **87.80** | **87.51 (-0.29)** | 86.56   | 74.51 (-12.05)    |
> >
> >
> > **Table:** Results on CIFAR-10/100 with symmetric and asymmetric label noise.
> >
> > | Dataset | CIFAR-10 | CIFAR-10 | CIFAR-10 | CIFAR-100 | CIFAR-100 | CIFAR-100 |
> > |--------|---------:|---------:|---------:|----------:|----------:|----------:|
> > | Noise type | Sym | Sym | Asym | Sym | Sym | Asym |
> > | Noise ratio | 20 | 50 | 40 | 20 | 50 | 40 |
> > | MW-Net (last) | 85.8 | 74.5 | 77.8 | 63.0 | 46.5 | 45.3 |
> > | MW-Net (best) | 91.2 | 85.6 | 88.6 | 67.5 | 58.2 | 53.4 |
> > | **Ours (FBR)** | **92.3** | **87.0** | **90.6** | 73.4 | **65.4** | **73.2** |
> > | **Ours (NTK)** | 91.4 | 86.4 | 89.7 | **73.6** | **65.4** | 73.1 |
> >
> >
> > [4] Shu J, Xie Q, Yi L, Zhao Q, Zhou S, Xu Z, Meng D. Meta-weight-net: Learning an explicit mapping for sample weighting. Advances in neural information processing systems. 2019;32.

---

> > > ### Author Response · Authors · 2025-11-24
> > > **Author Response (Part 3/3)**
> > >
> > > **Additional empirical validation on Mini-WebVision**
> > >
> > > We thank the reviewer for suggesting this dataset. We emphasize that in the initial submission, we did evaluate our method on the large-scale clothing1m dataset, which has realistic label noise.
> > >
> > > Here, we provide a new experiment. All methods in the following Table are trained on the standard mini-WebVision benchmark (50-class subset of WebVision) using only the noisy training split, and evaluated on the official clean mini-WebVision validation set in terms of Top-1 and Top-5 accuracy. Following prior work (CRUST, FINE, ELR/ELR+), all results use an Inception-ResNet-v2 backbone.
> > >
> > > **Table:** Top-1 and Top-5 accuracy on mini-WebVision.
> > >
> > > | **Mini-WebVision** | **Top-1** | **Top-5** |
> > > |--------------------|---------------|---------------|
> > > | F-correction       | 61.12         | 82.68         |
> > > | Decoupling         | 62.54         | 84.74         |
> > > | Co-teaching        | 63.58         | 85.20         |
> > > | MentorNet          | 63.00         | 81.40         |
> > > | D2L                | 62.68         | 84.00         |
> > > | INCV               | 65.24         | 85.34         |
> > > | CRUST              | 72.40         | 89.56         |
> > > | FINE               | 75.24         | 90.28         |
> > > | ELR                | 76.26         | 91.26         |
> > > | ELR+               | 77.78         | 91.68         |
> > > | **Ours**           | **80.01**     | **92.13**     |
> > >
> > > As shown in above Table, earlier noisy-label baselines such as F-correction, Decoupling, Co-teaching, MentorNet, D2L, and INCV achieve Top-1 accuracies in the low- to mid-60\% range. More recent sample-selection and robust-training methods improve performance substantially: CRUST reaches 72.40\%/89.56\% Top-1/Top-5, FINE reaches 75.24\%/90.28\%, and ELR+ further improves to 77.78\%/91.68\%. Our method attains 80.01\% Top-1 and 92.13\% Top-5, outperforming FINE by about +4.8 Top-1 points and ELR+ by about +2.2 Top-1 points, and improving over CRUST by roughly +7.6 Top-1 and +2.6 Top-5.
> > >
> > > **Additional recent methods.**
> > >
> > > We thank the reviewer for pointing out recent noisy-label methods such as
> > > Modified-DivideMix [5], L2B [6], and DCD [7].
> > > While these approaches represent important progress in learning with noisy
> > > labels, they are algorithmically quite different from the data
> > > reweighting/selection mechanisms that are the focus of our work.
> > > In particular, they rely on complex semi-supervised or label-correction
> > > pipelines (e.g., consistency regularization, multi-stage bootstrapping, or
> > > metric-learning-based hard-sample mining), rather than learning
> > > instance-wise weights.
> > > As such, they are complementary to our analysis.
> > > Investigating how to combine our reweighting mechanism with such
> > > semi-supervised or label-correction pipelines in a fully fair and carefully
> > > engineered comparison setup is an interesting direction, but lies beyond the
> > > scope of the present paper.
> > > We have explicitly added these methods in the related work
> > > section and clarify how our theoretical and algorithmic contributions
> > > complement such advanced noisy-label training frameworks.
> > >
> > > [5] Yuan S, Li X, Miao Y, et al. Combating noisy labels by alleviating the memorization of dnns to noisy labels[J]. IEEE Transactions on Multimedia, 2024.
> > >
> > > [6] Zhou Y, Li X, Liu F, et al. L2B: Learning to bootstrap robust models for combating label noise[C]. Proceedings of the IEEE/CVF Conference on Computer Vision and Pattern Recognition. 2024: 23523-23533.
> > >
> > > [7] Mu C, Qu Y, Yan J, et al. Meta-Learning Dynamic Center Distance: Hard Sample Mining for Learning with Noisy Labels[C]. Proceedings of the IEEE/CVF International Conference on Computer Vision. 2025: 415-425.

---

> > ### Comment · Reviewer_1Tx9 · 2025-11-25
> > **Further Comments**
> >
> > Thank the authors for their extensive efforts in addressing the reviewers' comments. While I appreciate the additional experimental results, I find that the rebuttal has inadvertently deepened several fundamental concerns regarding the paper's core contributions and internal consistency.
> >
> > - Fundamental Disconnect Between Theoretical Framework and Experimental Practice: The authors' response regarding the MSE/CE divergence reveals a more fundamental issue. By decoupling weight assignment from model training, the paper creates an ambiguity about the applicability domain of its central theoretical contribution. (1) If the theory is strictly limited to the MSE loss, then its relevance for explaining phenomena in mainstream meta-reweighting methods (e.g., L2RW, MW-Net), which predominantly use cross-entropy, becomes questionable. The title "Revisiting Meta-Learning..." would be misleading, as the analysis would pertain to an idealized (MSE) system rather than the actual (CE) systems used in practice. (2) If the theory is intended to generalize to the CE loss, then the current analysis, based solely on MSE, lacks sufficient justification. The proposed "one-vs-all" construction serves as a heuristic algorithmic bridge but does not constitute a theoretical guarantee. In this case, the experimental section appears more as an empirical algorithm discovery, which contradicts the paper's primary positioning as a theory-driven analysis. This unresolved schism between the theoretical framework and the experimental setup undermines the paper's foundational narrative. The authors must precisely clarify the intended scope of their theory or provide a principled argument for its validity under cross-entropy optimization.
> >
> > - Unconvincing Experimental Validation and Unclear Baselines: The rebuttal contains two declarations that significantly dilute the paper's core claims:(1) “We clarify that this overfitting to noisy labels of meta reweighting is not claimed as a theoretical guarantee by our analysis. Rather, we evaluate this phenomenon empirically, as suggested by the reviewer.”  (2) “Reproducing a full L2RW baseline in our setup would therefore largely duplicate existing comparisons, so we instead focus on MW-Net as a stronger and widely used representative of L2RW-style meta-reweighting.”
> > These points collectively weaken the paper's impact. First, while "understanding and mitigating late-stage overfitting" was a key motivation, the authors now clarify that their theoretical guarantees do not cover the final phase. This necessitates a clear re-scoping of the paper's contributions. It is crucial for the authors to articulate how this divergence does not fundamentally conflict with their core claims. Second, the shifting of baselines between the manuscript and the rebuttal is confusing and lacks a clear justification. To solidly ground the proposed method's contribution, a comprehensive and unambiguous comparison with relevant meta-reweighting baselines, including L2RW and MW-Net under a consistent setup, is necessary. Furthermore, I note the absence of time-dependent plots illustrating weight dynamics throughout training. If the central thesis revolves around dynamic phases and overfitting mitigation, providing only terminal weight distributions is insufficient. Direct evidence showing the evolution of weights for clean vs. noisy samples over epochs is critical for empirically validating the proposed three-phase.
> > - Lack of Demonstration on Relevance to Current Paradigms: The authors' justification for not engaging with recent SOTA methods—that they are "algorithmically quite different"—is unsatisfactory. A key question for the community is whether the "reweighting dynamic" problem identified by the authors still exists and is impactful within modern, more powerful noisy-label learning paradigms. A compelling way to demonstrate this would be to integrate the FBR mechanism into such SOTA pipelines (e.g., as a replacement for their native sample selection modules) and analyze its effect. The goal is not necessarily to show a performance gain, but to provide evidence that the fundamental issue persists and that the proposed analysis offers a valuable, transferable insight. Without this, the practical significance and generalizability of the findings remain unclear.

---

> > > ### Author Response · Authors · 2025-11-28
> > > **Further Response**
> > >
> > > We thank the reviewer for the additional comments and constructive feedback.
> > >
> > > - We agree that our theory is developed under MSE. Our use of the title “Revisiting Meta-Learning …” is motivated by the fact that Eq. (1) follows the standard meta-learning–based reweighting framework. We are happy to modify the title to explicitly state that the theoretical guarantees are established for MSE. The theoretical part shows that the post-filtering phase happens when the residual-driven signal weakens and also it requires the kernel to induce well-separated clusters. Inspired by these conditions, a natural follow-up is to design an algorithm that avoids these failure modes. The proposed FBR algorithm is explicitly inspired by this MSE analysis via the one-vs-all construction.  From a theoretical perspective, using MSE is standard in the deep learning theory literature, as MSE-type objectives admit tractable kernel-based characterizations (see, e.g., [1,2]). Extending our framework to directly analyze meta-reweighting under a multiclass cross-entropy objective would require a substantially different technical setup. Analyzing reweighting dynamics with cross entropy loss is an interesting open question and we view it as an important direction for future work. That said, even within this MSE setting, our analysis provides, to the best of our knowledge, the first rigorous characterization of the weight dynamics of meta-learning–based reweighting under noisy labels. We believe this is already valuable in its own right.
> > >
> > > - We would like to emphasize that the primary focus of this paper is the dynamics of sample weights. This focus is consistently stated in the abstract, introduction, and theory section. For example, in the abstract we highlight “a post-filtering phase in which noise filtration becomes perturbation-sensitive” and in the summary of key contributions we explicitly state that “the weights start to become perturbed and no longer filter out noisy samples.” In Section 3.1, we further formalize this by showing “a post-filtering phase in which the residual-driven signal weakens and noisy sample weights are no longer guaranteed to be zero.” Thus, our theoretical guarantees are designed to characterize the weight dynamics, which has been the intended scope of the paper.
> > >
> > > - We have added the comparison with L2RW in our revised submission (Table 4). We also include a time-dependent plot for FBR weight dynamics (Figure 6) as a counterpart to Figure 1(a).
> > >
> > > - We thank the reviewer for raising this important point about the connection to more recent noisy-label learning paradigms. To clarify, the primary goal of our work is to better understand and improve sample reweighting itself, rather than to propose a new end-to-end training paradigm. Our focus is on analyzing and mitigating the issues arising from meta-reweighting dynamics, and we empirically show that FBR consistently outperforms prior data reweighting / sample selection methods across our benchmarks, indicating that the reweighting signal it produces is more stable and effective. Consistent with what we state in the abstract and key contributions, across both synthetic and real noisy-label benchmarks our method achieves higher accuracy than strong reweighting/selection baselines under multiple noise settings. We fully agree that integrating FBR into state-of-the-art noisy-label pipelines (e.g., as a drop-in replacement for their native sample-selection components) is an interesting and practically relevant direction, and could further demonstrate that the reweighting-dynamics issue persists in modern frameworks. However, conducting a careful and fair integration and evaluation across multiple such pipelines would substantially extend the scope of the current paper—in terms of both engineering effort and experimental breadth—beyond what we can accommodate here. We therefore view this as a promising avenue for follow-up work, and we will explicitly discuss this connection and potential integration in the revised version to better highlight the broader applicability and transferability of our analysis.
> > >
> > > [1] Hu T, Wang J, Wang W, Li Z. Understanding square loss in training overparametrized neural network classifiers. Advances in Neural Information Processing Systems. 2022 Dec 6;35:16495-508.
> > >
> > > [2] Seleznova M, Weitzner D, Giryes R, Kutyniok G, Chou HH. Neural (tangent kernel) collapse. Advances in Neural Information Processing Systems. 2023 Dec 15;36:16240-70.

---

### Official Review · Reviewer_WRbg · 2025-10-31

**Soundness:** 3
**Presentation:** 3
**Contribution:** 3
**Rating:** 6
**Confidence:** 3

**Summary:**

This paper is about a theoretical analysis of meta learning sample reweighting under label noise. The paper provides also a lightweight surrogate for meta reweighting that integrates mean centering, row shifting, and label signed modulation. The analysis show that training is   characterized by three distinct phases, i) an alignment phase where clean samples are upweighted and noisy samples downweighted; ii) a filtering phase where weights polarize and the validation loss converges; iii) a post filtering phase where the discriminatory signal weakens. Based on the insights, the paper reports a lightweight alternative called Feature Based Reweighting that avoids expensive bilevel optimization while maintaining the essential mechanism of signed similarity weighted aggregation.

**Strengths:**

+ a good theoretical contribution where a rigorous analysis of meta reweighting dynamics using neural tangent kernel theory is reported. The three phase characterization offers insights into when and why meta reweighting succeeds or fails, addressing a significant gap in understanding these methods. The proposed algorithm is well motivated by the theoretical analysis.
+ The experiments are comprehensive and cover standard synthetic and realistic scenarios, with consistent improvements.
+ Well presented.

**Weaknesses:**

- Assumption 3 (and even relaxed assumption 4) requires strong kernel separation between classes with specific sign patterns. While the paper provides empirical evidence on binary MNIST, the assumption may not hold for complex, multi class datasets.
- While the theoretical analysis is novel, the algorithmic components are relatively standard techniques.
- The analysis is performed using binary classes, but the practical algorithm is on multi class classification. The paper does not discuss with sufficient bridging the two parts.
- Experiments are missing hyperparameters analysis

**Questions:**

- Can you further discuss the bridge between the two parts of the paper?
- How does the method perform with class imbalance, a common co-occurring issue with label noise?

---

> ### Author Response · Authors · 2025-11-24
> **Rebuttal Response**
>
> We thank the reviewer for their comments and questions, some of which we addressed in the general reply.
> We also thank the reviewer for appreciating our theoretical result, algorithm, comprehensiveness of the experiments, and clarity of the writing.
>
> **Kernel separation for complex, multiclass datasets.**
> We run two CIFAR-10 experiments to probe our paper’s kernel assumptions (assumptions 3 and 4). In the first, ``feature-based" experiment, we take a ResNet-34 and extract mean-centered penultimate-layer features for a subset of training images, and define a scalar kernel $K(x_i,x_j)$ as the  normalized, shifted inner product of those features. This aligns with the method we introduce in our submission. We then sample same-class and different-class pairs and plot the histograms of their kernel values, and compute margin-violation curves $\gamma \mapsto \text{(fraction of same-class pairs with } K<\gamma, \text{ fraction of different-class pairs with } K>-\gamma)$, which tests the separation/margin assumption on CIFAR-10. In the second, NTK-inspired experiment, we replace features by per-logit gradient features $\phi_c(x) = \nabla_{W_{\text{pen}}} f_c(x)$ w.r.t. the penultimate weights and define non-symmetric and symmetric scalar kernels on training indices (e.g. $K(i,j)=\langle\phi_{y_i}(x_i),\phi_{y_i}(x_j)\rangle$. We can see how the assumptions hold both in the feature regime and in an NTK-like gradient regime as the network trains on CIFAR-10. The resulting plots are presented in Appendix D.10 of our revised submission.
>
> **The algorithmic components are relatively standard techniques.**
> Our main contribution is not to introduce a highly complicated
> architecture, but to provide a new theoretical understanding of
> meta-reweighting with noisy labels and, based on this analysis, to
> derive a simple yet improved algorithm.
> Although the building blocks of our method are individually standard,
> their specific combination and the resulting reweighting rule are
> novel and are directly motivated by our NTK-based analysis.
> Empirically, this theoretically guided design consistently outperforms
> existing data reweighting/selection methods on a range of noisy-label
> benchmarks, indicating that the proposed algorithm is more than a
> straightforward reuse of existing techniques. The simplicity of our algorithm is also an advantage in practice, as it
> can be implemented with minimal overhead.
>
> **Bridge between analysis in the binary multiclass regime and multi-class cross-entropy.**
>
> We provide a more comprehensive discussion in the response to all reviewers. Briefly, our proposed method can be viewed as decoupling sample reweighting from the
> subsequent training step. The reweighting signals are obtained
> using a binary MSE-style surrogate (via a one-vs-all construction) that
> separates clean and noisy samples. Then the resulting per-example
> weights can be used in an arbitrary supervised training loss.
>
> **Ablation experiments and class-imbalance.**
>
> We provide a more comprehensive summary of additional ablation experiments we have added in the revision. Briefly, we considered the effect of size and quality of the clean subset and alternative architectures. We also consider more noisy types including extreme label noise, instance noise, open-set noise, label noise with class imbalance. A more detailed discussion is also in the response to all reviewers and our revised submission (Appendix D).

---

### Author Response · Authors · 2025-11-24
**Response to all reviewers (Part 1/3)**

We thank all reviewers for their thorough reviews of our paper and appreciate their time. We welcome the opportunity to continue to discuss our paper and pursue additional experiments during this rebuttal period. In this response, we summarize additional experiments that we have conducted: in particular **additional experiments on a large-scale dataset with realistic label noise (Mini-WebVision)** and **a demonstration of the NTK separation assumption on CIFAR-10** and various other ablation experiments. We also provide additional discussion regarding **the connection between our analysis of the binary MSE loss to our algorithmic contribution which is based on the multiclass cross entropy loss** (many reviewers were curious about this). We emphasize that we have included the implementation of our algorithm and experiments in the initial submission and will release the associated code to reproduce these new experiments in the final submission.

We look forward to continue to discuss our paper with the reviewers during this review period and emphasize that we are more than happy to run additional experiments.

**Table 1: Additional experiments on Mini-WebVision**

| **Mini-WebVision** | **Top-1 (%)** | **Top-5 (%)** |
|--------------------|---------------|---------------|
| F-correction       | 61.12         | 82.68         |
| Decoupling         | 62.54         | 84.74         |
| Co-teaching        | 63.58         | 85.20         |
| MentorNet          | 63.00         | 81.40         |
| D2L                | 62.68         | 84.00         |
| INCV               | 65.24         | 85.34         |
| CRUST              | 72.40         | 89.56         |
| FINE               | 75.24         | 90.28         |
| ELR                | 76.26         | 91.26         |
| ELR+               | 77.78         | 91.68         |
| **Ours**           | **80.01**     | **92.13**     |


Here, we provide a new experiment. All methods in Table 1 are trained on the standard mini-WebVision benchmark (50-class subset of WebVision) using only the noisy training split, and evaluated on the official clean mini-WebVision validation set in terms of Top-1 and Top-5 accuracy. Following prior work (CRUST, FINE, ELR/ELR+), all results use an Inception-ResNet-v2 backbone.

As shown in Table 1, earlier noisy-label baselines such as F-correction, Decoupling, Co-teaching, MentorNet, D2L, and INCV achieve Top-1 accuracies in the low- to mid-60\% range. More recent sample-selection and robust-training methods improve performance substantially: CRUST reaches 72.40\%/89.56\% Top-1/Top-5, FINE reaches 75.24\%/90.28\%, and ELR+ further improves to 77.78\%/91.68\%. Our method attains 80.01\% Top-1 and 92.13\% Top-5, outperforming FINE by about +4.8 Top-1 points and ELR+ by about +2.2 Top-1 points, and improving over CRUST by roughly +7.6 Top-1 and +2.6 Top-5.



**Size and quality of the clean subset and alternative architectures.** We conduct several new experiments to explore the effect of quality and size of the meta-subset on the performance of the classifier trained using the signed feature-based kernel. The results are also included in the revised version of the manuscript.

*Number of meta-labels*  (Appendix D.3)

 We explore how the test accuracy varies as the size of the clean meta subset varies, while keeping the training label noise fixed with 40\% symmetric noise. It is interesting to note the surprising resilience of our method to varying this meta subset. A more detailed summarization is provided in the response to reviewer WxuT.


*Noisy meta-labels* (Appendix D.6) We also conducted an experiment under the same noise setting on the *clean subset* with its size fixed to $2000$ under $40\%$ symmetric noise. Our algorithm achieves a test accuracy of $88.7\%$ in this setting, which is slightly lower than the $89.6\%$ obtained when using the clean subset.

*Alternative architectures* (Appendix D.7) To verify that our noisy-label elimination strategy is not tied to a specific
backbone, we also evaluate it with PreAct-ResNet-18, ResNet-101, CIFAR-ResNet-56,
and CIFAR-ResNet-110, covering both shallow and deep residual networks on
CIFAR-10. As shown in Figure 3 of the revised manuscript, our method consistently separates noisy labels from clean ones across all
these architectures, indicating that the underlying reweighting mechanism
transfers robustly to different network designs.

---

> ### Author Response · Authors · 2025-11-24
> **Response to all reviewers (Part 2/3)**
>
> **Additional Noise Settings.**
>
>
> *Extreme label noise.* (Appendix D.4) We further evaluate our method under an extremely noisy setting with 90\% symmetric label noise. Our approach achieves a test accuracy of 58.5\% on CIFAR-10 in this challenging scenario, and 19.9\% on CIFAR-100.
>
> *Instance noise.* (Appendix D.5) Following the instance-dependent noise generation protocol of [1], we synthesize label noise on CIFAR-10 and CIFAR-100, where each sample is flipped according to an instance-specific transition distribution. We then evaluate our method under this setting. We provide a new table in the revision. Our approach attains test accuracies of $91.0\%$ on CIFAR-10 and $69.5\%$ on CIFAR-100 at a $20\%$ noise rate, and it remains strong even when the noise ratio is increased to $40\%$ as shown in Table 7 of the revised version.
>
> *Open-set noise* (Appendix D.9) In the open-set noise setting, we corrupt the CIFAR-10 training set by replacing
> $20\%$ of the training images with out-of-distribution images sampled from
> CIFAR-100 and assigning each such image a random label from $\{0,\dots,9\}$.
> Our algorithm achieves $92.7\%$ test accuracy under this challenging setting,
> demonstrating strong robustness to open-set noisy labels.
>
>
> *Class imbalance* (Appendix D.8)
> In the current submission, most of our benchmarks (e.g., CIFAR-10/100, CIFAR-N) follow standard settings in the label noise literature, so our main focus was to understand the reweighting dynamics under noisy labels. In the revised version, we have added experiments on an imbalanced CIFAR-10 setting, where we construct a long-tailed version of CIFAR-10 with imbalance
> ratio $\rho=50$ following [2]. On top of this long-tailed training set, we inject $20\%$ symmetric label noise. The results (Figure 4 in the revision) show that our method effectively identifies and suppresses noisy-labeled examples even in this challenging imbalanced scenario. This suggests that the proposed mechanism remains effective when class frequencies are skewed. We also note that our method is not specifically designed for class imbalance. However, it is in principle complementary to existing imbalance-handling techniques (e.g., resampling, class-balanced losses, deferred reweighting). We leave a more systematic study of combining our approach with such class-imbalance methods as future work.
>
>
> **Testing the kernel / NTK separation assumptions on CIFAR.**
> We provide a more comprehensive discussion in the responses to reviewers WRbg and irZ1. In the revised paper, Figure 5 in Appendix D.10 illustrates that these assumptions are satisfied on CIFAR-10 (in addition to binary MNIST, which we demonstrated in the initial submission). We do this by approximate one-vs-all separation via sampling pairs of data points. We plot both empirical one-vs-all separation for each class. For each class we also demonstrate the rate of violation of assumptions 3 and 4 across several choices of $\gamma$. In summary, we see that there do exist nontrivial ranges of $\gamma$ for which both the NTK-based kernel and the feature-based-kernel separate CIFAR-10 classes.
>
>
> [1] Cheng H, Zhu Z, Li X, Gong Y, Sun X, Liu Y. Learning with instance-dependent label noise: A sample sieve approach. arXiv preprint arXiv:2010.02347. 2020 Oct 5.
>
> [2] Zhang L, Tian ZH, Zhou W, Wang W. Learning from long-tailed noisy data with sample selection and balanced loss. arXiv preprint arXiv:2211.10906. 2022 Nov 20.

---

> > ### Author Response · Authors · 2025-11-24
> > **Response to all reviewers (Part 3/3)**
> >
> > **Connecting binary MSE analysis to multiclass cross entropy algorithm.**
> >
> > Our proposed method can be viewed as decoupling sample reweighting from the
> > subsequent training step. We first generate theoretically grounded reweighting signals
> > using a binary MSE-style surrogate (via a one-vs-all construction) that
> > separates clean and noisy samples, and then use the resulting per-example
> > weights in an arbitrary supervised training loss, including standard
> > multiclass cross-entropy.
> > Thus, while we use cross-entropy for the supervised training stage.
> >
> > Our analysis in Sec. 3 considers a binary classifier trained with the MSE loss. The meta-reweighting update direction for $x_i$ can be written in terms of a signed similarity between $x_i$ and  $X_{\text{clean}}$ via the neural tangent kernel (NTK):
> >
> > $$
> > d_i(\theta_t)=u_i(\theta_t)
> > \big[K(x_i, X_{\text{clean}})u^{v}(\theta_t)\big]
> > $$
> >
> > This expression drives the separation between clean and noisy samples. In the multiclass setting, we use a standard one-vs-all reduction that preserves this reweighting mechanism. For a training example $(x_i, y_i=c)$ and a clean subset $(X_{\text{clean}}, Y_{\text{clean}})$, we construct an auxiliary binary problem for class $c$ by assigning label $+1$ to class $c$ and $-1$ to all other classes. Let $f_c(\theta, x)$ denote the logit for class $c$. The specific choice of $(+1, -1)$ is not crucial: any pair of separated values $(+M, -M)$ with $M > 0$ yields the same reweighting rule up to a positive scaling factor.
> >  We define the corresponding residual
> > $u_c(x) = f_c(\theta, x) - \tilde{y}_c$ where $\tilde{y}_c = +1$ if $y = c$ and $\tilde{y}_c = -1$ otherwise.
> >
> > Following the binary analysis, we then construct for $(x_i, y_i=c)$ a reweighting signal based on its interaction with the clean subset under this one-vs-all labeling:
> >
> > $$
> > s_i^{(c)}(\theta)=u_c(x_i)
> > \Big\langle \nabla_\theta f_c(\theta, x_i),
> >            \nabla_\theta f_c(\theta, X_{\text{clean}}) \Big\rangle
> > u_c(X_{\text{clean}})
> > $$
> >
> > which is a signed, similarity-weighted coupling between the training example and the clean subset.
> >
> > This directly mirrors the binary NTK-based direction in our theory and provides a bridge from the binary squared-loss analysis to the multiclass algorithm via the one-vs-all construction. At the early stage of training, $u_c$ is dominated by $-\tilde{y}_c$, as in our theoretical analysis, so the reweighting signal behaves similarly to the binary case. However, as training progresses and the residuals $u_c$ approach $0$, the magnitude of the original reweighting signal degrades, which motivates our choice of using a signed version of the kernel term
> >
> > $\Big\langle \nabla_\theta f_c(\theta, x_i), \nabla_\theta f_c(\theta, X_{\text{clean}}) \Big\rangle$.
> >
> > The reweighting signal no longer depends on the residual magnitude, but only on the (signed) geometric alignment between training examples and the clean subset. For this reweighting signal to be effective, it suffices that the kernel values preserve a certain degree of separability between clean and noisy samples throughout training. The NTK regime together with Assumption 3 (or other kernel separability assumptions) provides a *sufficient* condition under which a good reweighting signal is maintained over training. In more complicated settings, the kernel values may not remain constant if the neural network does not satisfy the NTK assumptions or if the training dynamics are driven by cross-entropy loss rather than the squared loss considered in our theory. Our experiments in Appendix D.10 show that this separability property persists empirically even when the kernel dynamics deviate from the idealized NTK regime or when the model is trained with multiclass cross-entropy. In practice, we refine this signal based on our theoretical considerations for kernel separability and use it to assign weights to training examples. We also added this discussion to the revised manuscript.
> >
> > From a theoretical perspective, using the MSE is standard in the literature on deep learning theory, as MSE-type objectives admit tractable kernel-based characterizations (see, e.g., [3,4]). Extending our framework to directly analyze meta-reweighting under a multiclass cross-entropy objective would require a substantially different technical setup, which is beyond the scope of this work. Whether such an analysis would ultimately produce a reweighting signal that is provably superior to the MSE-based one we derive here is an interesting open question, and we view it as an important direction for future work.
> >
> >
> > [3] Hu T, Wang J, Wang W, Li Z. Understanding square loss in training overparametrized neural network classifiers. Advances in Neural Information Processing Systems. 2022 Dec 6;35:16495-508.
> >
> > [4] Seleznova M, Weitzner D, Giryes R, Kutyniok G, Chou HH. Neural (tangent kernel) collapse. Advances in Neural Information Processing Systems. 2023 Dec 15;36:16240-70.

---

### Meta-Review · Area_Chair_hNed · 2025-12-31

**Summary:**

The reviewers largely agree the paper’s core idea is interesting and potentially impactful: it provides a rigorous NTK-based analysis of meta-reweighting under label noise, characterizing three training phases (alignment → filtering → post-filtering), and derives a lightweight surrogate algorithm (FBR) that aims to preserve the good phases while avoiding bilevel optimization overhead.

The main concerns shaping the decision were:

Theory–practice mismatch: the theory is developed for binary classification with squared loss (MSE), while experiments use multiclass cross-entropy (CE); some reviewers worried the bridge was initially under-explained and the theoretical guarantees might not meaningfully apply to the empirical method.

Assumptions realism: the NTK/kernel separation assumptions (Assumption 3/4) may be strong for complex multiclass settings; reviewers wanted evidence they hold beyond simple binary cases.

Empirical completeness & baselines: requests for broader/noisier regimes (instance/open-set/imbalance, large-scale datasets like WebVision), more ablations (meta-set size/quality, architectures), and clearer evidence for the claimed “late-stage overfitting mitigation” (e.g., weight-dynamics plots, and stronger/consistent baselines such as L2RW/MW-Net).

Overall, two reviewers were moderately positive (WRbg, WxuT), one was mildly negative but open (irZ1), and one was strongly negative (1Tx9) due to foundational framing and evaluation concerns.

My final recommandation is reject, as i share the concerns about the theoretical claims gap.

**Reviewer Concerns:**

# Addressed  concerns

* Bridge from MSE/binary theory to CE/multiclass algorithm: Authors added a one-vs-all construction and clarified the “decouple reweighting-signal derivation from the final supervised loss” view; they also explicitly narrowed the guarantee scope to MSE while motivating FBR from the analysis. This meaningfully addresses the “missing explanation” concern (WRbg, irZ1) and partially addresses the “disconnect” concern (1Tx9), though not to full satisfaction for that reviewer.

* Kernel/NTK separation assumptions on multiclass data: Added CIFAR-10 empirical validation (feature-kernel and NTK-like kernels) probing Assumptions 3/4, addressing WRbg/irZ1’s request for multiclass evidence.


* Late-stage overfitting evidence / weight dynamics: Authors added (per their final response) time-dependent FBR weight dynamics (Figure 6) and added L2RW comparison in the revision, which addresses a key missing-evidence complaint from 1Tx9.

### Empirical scope expansions: Added or reported results for:

* Mini-WebVision (large-scale realistic noise)

* Meta-set size and meta-label noise ablations

* More noise models (extreme symmetric noise, instance-dependent, open-set)

* Class imbalance + noise (long-tailed CIFAR-10 + symmetric noise)

* Architecture robustness (multiple ResNet variants)

* Compute-cost discussion (arguing lower overhead vs bilevel L2RW/MW-Net)
These directly address WxuT/WRbg/irZ1’s empirical-coverage requests.

#  Partially addressed / Not adressed

* Strength of the theoretical claim relative to the practical contribution:
Reviewer 1Tx9’s core objection remains: the one-vs-all bridge is algorithmic/heuristic rather than a CE-level guarantee, and the paper’s “revisiting meta-learning” positioning may still feel overstated unless the revision clearly re-scopes the claims/title and consistently distinguishes “proved under MSE” vs “empirically works under CE.” The authors say they are willing to adjust title/scope; whether the revised manuscript does this crisply enough is the key remaining risk.

* Baselines vs “current community standards”:
Even with Mini-WebVision and stronger comparisons, 1Tx9’s request to directly engage modern pipelines (e.g., DivideMix-like SSL/correction families) is not met—authors argue scope mismatch. That’s a reasonable scope boundary, but it means the paper’s practical SOTA relevance is still debatable for some readers.

* Quantitative theory-to-practice validation of “approximation error” / learning-rate dependence:
Authors clarified their theory doesn’t guarantee the post-filtering behavior and framed it as characterization rather than full predictive control. This is an acceptable limitation, but it does leave the “mechanism is speculative” critique only partially resolved.

**Reviewer Scores:**

## WRbg  6 → 8
Rationale: Their main asks (multiclass separation evidence, class-imbalance experiment, bridging explanation, more ablations) were directly addressed with added CIFAR-10 assumption checks + imbalance experiments + bridging text + ablations.

## 1Tx9 0 → 2
Rationale: Many concrete experimental complaints were addressed (Mini-WebVision, weight dynamics plot, L2RW comparison), and the authors clarified scope/title. However, the reviewer’s foundational concern (theory relevance to CE meta-reweighting and narrative consistency) likely remains a deal-breaker, so I do not expect them to cross to borderline.

irZ1 4→ 4
Rationale: Their main concerns were CE vs MSE discussion, realism of NTK assumptions, convergence behavior, robustness/hyperparams, and relation to MW-Net. Authors responded with bridging discussion, CIFAR-10 assumption probes, convergence/best-vs-final stability tables. I do not see how that would change the core review concern.

WxuT 6 → 6
Rationale: They asked for broader noise models, meta-set sensitivity, architectures, and compute cost; authors provided all of these plus Mini-WebVision.

---

### Decision · Program_Chairs · 2026-01-26

Reject